# LESS IS MORE: TOWARDS SIMPLE GRAPH CONTRASTIVE LEARNING

## ABSTRACT

Graph Contrastive Learning (GCL) has shown strong promise for unsupervised graph representation learning, yet its effectiveness on heterophilic graphs, where connected nodes often belong to different classes, remains limited. Most existing methods rely on complex augmentation schemes, intricate encoders, or negative sampling, which raises the question of whether such complexity is truly necessary in this challenging setting. In this work, we revisit the foundations of supervised and unsupervised learning on graphs and uncover a simple yet effective principle for GCL: mitigating node feature noise by aggregating it with structural features derived from the graph topology. This observation suggests that the original node features and the graph structure naturally provide two complementary views for contrastive learning. Building on this insight, we propose an embarrassingly simple GCL model that uses a GCN encoder to capture structural features and an MLP encoder to isolate node feature noise. Our design requires neither data augmentation nor negative sampling, yet achieves state-of-the-art results on heterophilic benchmarks with minimal computational and memory overhead, while also offering advantages in homophilic graphs in terms of complexity, scalability, and robustness. We provide theoretical justification for our approach and validate its effectiveness through extensive experiments, including robustness evaluations against both black-box and white-box adversarial attacks.

## 1 INTRODUCTION

Contrastive learning is a powerful unsupervised technique for representation learning that has attracted significant attention in recent years. It learns meaningful representations by encouraging embeddings of similar instances to align closely while pushing apart those of dissimilar ones, typically using feature embeddings generated from different encoders. This process allows models to capture important patterns without relying on large amounts of labeled data and has demonstrated strong performance in domains such as computer vision, natural language processing, and recommendation systems (Radford et al., 2021; Grill et al., 2020; Chen et al., 2020). When extended to graph-structured data, this approach is referred to as Graph Contrastive Learning (GCL).

The central idea of GCL is to design encoders that produce distinct yet semantically meaningful graph views. While this paradigm has shown strong promise, its effectiveness on heterophilic graphs, where connected nodes often belong to different classes, remains limited. To overcome this challenge, many existing frameworks adopt increasingly complex strategies. Augmentation-based approaches generate views through perturbations such as edge removal or feature masking (Zhang et al., 2023; Zhu et al., 2020; 2021; Xiao et al., 2022; Xu et al., 2025), often using elaborate, heuristically designed pipelines that may distort graph semantics. For example, EPAGCL (Xu et al., 2025) constructs augmented views by adding or dropping edges according to weights derived from the Error Passing Rate (EPR). In contrast, augmentation-free approaches shift the complexity to the encoder, requiring sophisticated designs to extract distinct representations from the same input. PolyGCL (Chen et al., 2024) applies polynomial filters to generate low-pass and high-pass spectral views, while SDMG (Zhu et al., 2025) employs two dedicated low-frequency encoders to facilitate diffusion-based learning. Despite these intricacies, both approaches often continue to rely on negative sampling during training, which adds further complexity. We refer readers to Appendix A for additional related work.

To illustrate the trade-off between complexity and node classification performance in recent GCL methods, Fig. 1 plots accuracy against training time for each epoch, with marker color indicating storage cost. They are on the more challenging heterophilic datasets: the Wisconsin dataset and the (large-scale) Roman dataset. GraphACL'23 and PolyGCL'24 gain higher accuracy at the expense of greater complexity, while GraphECL'24, EPAGCL'25, and SDMG'25 reduce training time and storage but suffer degraded performance. This trend raises two natural questions:

Q1  *Have recent advances in GCL substantially improved performance on heterophilic graphs?*

Q2  *Are increasingly elaborate designs truly necessary, or can simpler models achieve comparable or better results?*

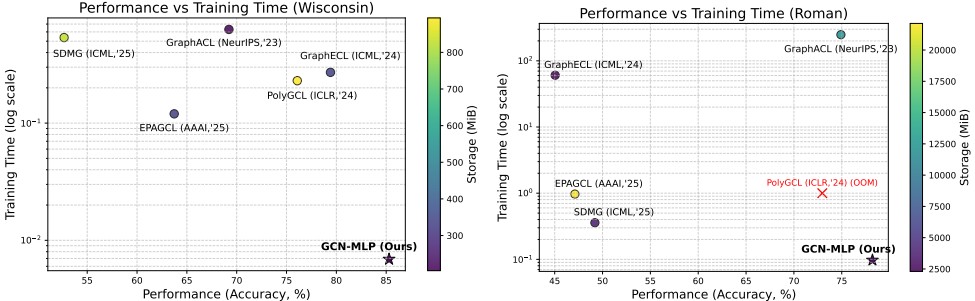

Figure 1: Performance–complexity trade-off of GCL methods on the Wisconsin and Roman datasets. Accuracy is plotted against training time (log scale, in seconds), with marker color indicating storage cost. Our GCN-MLP achieves the best performance with minimal complexity, while OOM cases are marked with red crosses. OOM refers to out of memory on an NVIDIA RTX A5000 GPU (24GB).

To address these questions, we revisit the essence of node classification. In the *ideal* case, classification is trivial when nodes of the same class share identical features. In practice, however, node features from the same class are better modeled as realizations from a common distribution. The *noise*, defined as the deviation of the feature from the distribution mean, introduces variability that complicates classification. Hence, we want to "mitigate noise", which might be (partially) achieved by aggregating features across nodes of the same class, akin to a law-of-large-numbers effect (Ji et al., 2025). In homophilic graphs, models such as GCN leverage neighborhood aggregation under the assumption that neighbors are likely to share the same class. In contrast, for heterophilic graphs, effective strategies involve identifying non-neighboring but same-class nodes for aggregation (Linkerhägner et al., 2025). This is usually much harder (Xiao et al., 2023), as the graph topology does not provide direct information for aggregation. While labeled data provides supervision to guide class separation, unsupervised settings require stronger noise mitigation to ensure that features cluster well by class. In summary, heterophilic GCL suffers from limited guidance from both the graph structure and node labels.

However, beyond aggregating features across nodes, an alternative way to mitigate noise is to generate multiple feature representations for the same node. Our strategy is motivated by the observation that: cancellation is stronger in the sum of two vectors when they are less correlated. The key, therefore, is to construct diverse feature views such that their associated "noise" is preferably less correlated. For graph-structured data, two natural sources arise: the original node features independent of the graph topology and the embeddings from aggregating over the graph structure. We hope that their respective noises, termed *feature noise* and *structural noise*, are weakly correlated for cancellation.

From the above intuition, an embarrassingly *simple* GCL model is readily available: we use *only* a *GCN* and an *MLP* as view-generation encoders. We emphasize that our novelty lies not in merely combining existing architectures (i.e., GCN and MLP), but in uncovering a novel underlying principle for GCL: when feature noise and structural noise are weakly correlated, their contrastive interaction (and simple linear fusion) yields stronger noise mitigation. Therefore, our design goal is to construct two views whose noise components are as uncorrelated as possible. The GCN-MLP architecture is a simple yet effective instantiation of this principle, where the GCN captures structural features together with their inherent structural noise, while the MLP isolates node feature noise, yielding two complementary views for contrastive learning. This GCN-MLP model requires neither data augmentation nor negative sampling, and it can be applied to any graph dataset. The approach has

notable advantages in heterophilic settings, where original features and graph structure are less correlated. We refer to it as "simple" due to its minimal and transparent design. As a preview, its effectiveness on certain datasets is demonstrated in Fig. 1, while more studies can be found in the main text.

Our main contributions are as follows:

- We propose an augmentation-free GCL model that is simple, flexible, and efficient. We provide the theoretical justification for our choice of contrasting views, which further explains the model's simplicity and robustness.
- We identify the reasons underlying the model's pronounced performance on heterophilic datasets, and we further demonstrate its cost-effectiveness, scalability, and strong robustness when applied to homophilic datasets.
- We conduct extensive numerical experiments on diverse datasets, showing clear advantages on many datasets in terms of accuracy, efficiency, and resistance to adversarial attack.

## 2 PRINCIPLES OF GRAPH CONTRASTIVE LEARNING

Consider an undirected graph $\mathcal{G} = (\mathcal{V}, \mathcal{E})$ with the node set $\mathcal{V} = \{v_1, \ldots, v_N\}$ and edge set $\mathcal{E} \subseteq \mathcal{V} \times \mathcal{V}$. Each node $v_i$ is associated with a feature vector $\mathbf{x}_i$, which is collected as the $i$-th row in the feature matrix $\mathbf{X} \in \mathbb{R}^{N \times d}$. The graph structure is encoded by a symmetric weighted matrix $\mathbf{A} = (a_{ij})_{1 \leq i,j \leq n} \in \mathbb{R}^{N \times N}$, where $a_{ij}$ is the edge weight between $v_i$ and $v_j$. The complete graph data is denoted by $\mathcal{X} = (\mathbf{A}, \mathbf{X})$.

GCL belongs to the category of *unsupervised representation learning*, where no labels are available during training. For unsupervised learning, the goal is to train an encoder $f_\theta$ that maps each node $v_i$ and its context in $\mathcal{X}$ to a representation $\mathbf{z}_i = f_\theta(\mathcal{X}, v_i) \in \mathbb{R}^F$, where $F$ denotes the feature dimension. The resulting embedding matrix $\mathbf{Z} \in \mathbb{R}^{N \times F}$ is then used for downstream tasks such as *node classification*, which is our main focus.

For GCL, self-supervision is achieved by enforcing consistency between representations $\mathbf{Z}_1$ and $\mathbf{Z}_2$ obtained from different encoders $f_{\theta_1}$ and $f_{\theta_2}$. As a guiding principle, the encoders $f_{\theta_1}$ and $f_{\theta_2}$ should represent different "graph views". The concept of a *graph view* is not universally defined and is open to interpretation, while a *local-global* dichotomy is popular (Chen et al., 2024). For each node $v_i$, the final feature representation is a weighted sum $\beta \mathbf{z}_{1,i} + (1 - \beta)\mathbf{z}_{2,i}$, where $\mathbf{z}_{1,i}$ and $\mathbf{z}_{2,i}$ are the $i$-th row of $\mathbf{Z}_1$ and $\mathbf{Z}_2$, respectively, and $0 < \beta < 1$. Most models simply choose $\beta = 0.5$ to avoid discriminating against any graph view.

We take a step back and examine the main challenge of unsupervised learning. In the ideal situation where the features are *noiseless*, i.e., $\mathbf{x}_i = \mathbf{x}_j$ if and only if $v_i$ and $v_j$ have the same label, the classification becomes trivial. However, this never happens for real datasets. More specifically, we formalize the discussion as follows.

Assume that for each label class $c$, the feature for a node with label $c$ is generated according to a class-specific distribution $\gamma_c$.

**Definition 1.** *Let $c$ be a class label and $\mathcal{V}_c$ be the set of nodes of label $c$. Define the* class centroid *$\mathbf{x}_c = \mathbb{E}_{\mathbf{x} \sim \gamma_c}[\mathbf{x}]$, and the* noise *of $v_i \in \mathcal{V}_c$ to be $\mathbf{n}_i = \mathbf{x}_i - \mathbf{x}_c$.*

In practice, a proxy for $\mathbf{x}_c$ is the empirical centroid $\widehat{\mathbf{x}}_c = (\sum_{v_i \in \mathcal{V}_c} \mathbf{x}_i)/|\mathcal{V}_c|$.

Unlike in the ideal situation, $\mathbf{n}_i$ can have a large norm, which prohibits effective separation of nodes from different classes. To address this challenge, we may aim for a small ratio between the norms of the noise and class centroid, termed *noise-to-class centroid ratio* (NCR), which leads to the following two natural strategies (see more discussions at the end of the section):

- Enlarge the norm of the class centroids of the output representation.
- Reduce the norm of the noise of the output representation.

To explain how these strategies might be implemented, we consider the following simple observation (see Appendix B for the proof).

**Proposition 1.** *Consider two different representation learning models $f_{\theta_1}$ and $f_{\theta_2}$ whose representations are in the same space $\mathbb{R}^F$. Let $\mathbf{z}_{1,c}$ and $\mathbf{z}_{2,c}$ be the respective centroids from these two different representations, assumed to be non-zero vectors. Then, for $0 < \beta < 1$, the norm of the aggregated feature centroid $\mathbf{z}_c = \beta\mathbf{z}_{1,c} + (1-\beta)\mathbf{z}_{2,c}$ increases as the cosine similarity between $\mathbf{z}_{1,c}$ and $\mathbf{z}_{2,c}$ increases while keeping the centroid norms fixed. Moreover, suppose $|\mathcal{V}_c| = n_c$, and for $r = 1, 2$, let $\mathbf{z}'_{r,c}$ be the empirical centroid of the features $\{\mathbf{z}_{r,i} : 1 < i \leq n_c\}$ (i.e., excluding the node $v_1$). Let $\mathbf{n}'_{r,1} = \mathbf{z}_{r,1} - \mathbf{z}'_{r,c}$ be the deviation of $v_1$'s feature from the empirical centroid in the $k$-th representation. Then, the norm of the output noise $\mathbf{n}_1 = \mathbf{z}_1 - \mathbf{z}_c$, where $\mathbf{z}_1 = \beta\mathbf{z}_{1,1} + (1-\beta)\mathbf{z}_{2,1}$, is a non-decreasing function of the cosine similarity between $\mathbf{n}'_{1,1}$ and $\mathbf{n}'_{2,1}$ while keeping the deviation norms fixed.*

Intuitively, recall that we seek cancellation between $\mathbf{n}_{1,i}$ and $\mathbf{n}_{2,i}$, while avoiding it between $\mathbf{z}_{1,i}$ and $\mathbf{z}_{2,i}$. The observation suggests that to generate output features $\mathbf{Z}$ with a small NCR, we need to ensure that for each $v_i$, $\mathbf{z}_{1,i}$ and $\mathbf{z}_{2,i}$ are strongly correlated, while their respective noise (to centroid) $\mathbf{n}_{1,i}$ and $\mathbf{n}_{2,i}$ are weakly correlated (see empirical evidence in Appendix D.1).

**On implementing the strategies** Contrastive learning is deemed to amplify the class centroid via a contrastive loss (minimizing pairwise feature cosine similarity). More specifically, the learning process seeks to align the centroids of distinct views. The dedicated loss encourages these centroids to form a small angle, thereby reducing cancellation during aggregation.

However, a similar approach to reducing noise through a dedicated loss is not as straightforward, since labels are unavailable during training. Centroids, and hence noise, cannot be computed explicitly. Instead, the idea is to design encoders with different characteristics so that their respective noise is intrinsically less correlated. We provide the motivation in the next section.

## 3 FEATURE NOISE AND STRUCTURAL NOISE

As we have envisioned in the previous section, we motivate the model design aiming for noise reduction. Any graph dataset naturally consists of two pieces of information: *features* and the *graph structure*. We formalize earlier discussions and associate each with a notion of "noise". We analyze their correlations, in alignment with the objective of noisy reduction as discussed in the previous section. Let $\widetilde{\mathbf{A}}_{\mathcal{G}}$ be the normalized adjacency matrix. For a matrix $\mathbf{M}$, we use $\mathbf{M}_i$ to denote the $i$-th row vector of $\mathbf{M}$.

**Definition 2.** *For any feature matrix $\mathbf{X}$, its associated* feature noise *of a node $v_i$ with label $c$ is $\mathbf{n}_i = \mathbf{x}_i - \mathbf{x}_c$. For a fixed $k > 0$, the $k$-hop* structural noise *(or simply the* structural noise*) of $v_i$ with class label $c$ is the feature noise $\mathbf{n}_i^{(k)}$ associated with the transformed feature matrix $\widetilde{\mathbf{A}}_{\mathcal{G}}^k\mathbf{X}$, defined as follows:*

$$\mathbf{n}_i^{(k)} = \left(\widetilde{\mathbf{A}}_{\mathcal{G}}^k\mathbf{X}\right)_i - \mathbf{M}_i \tag{1}$$

*with*

$$\mathbf{M}_i = \mathbb{E}\left[\frac{1}{|\mathcal{V}_c|}\sum_{v_j \in \mathcal{V}_c}\left(\widetilde{\mathbf{A}}_{\mathcal{G}}^k\mathbf{X}\right)_j\right] = \frac{1}{|\mathcal{V}_c|}\sum_{v_j \in \mathcal{V}_c}\left(\widetilde{\mathbf{A}}_{\mathcal{G}}^k\overline{\mathbf{X}}\right)_j, \tag{2}$$

*where $\overline{\mathbf{X}}$ is the mean feature matrix whose $j$-th row is $\mathbf{x}_c$ if $v_j$ has label $c$.*

Observe that when $k = 0$, the 0-hop structural noise reduces to ordinary feature noise associated with $\mathbf{X}$. We single out this case since no graph information is involved. Recall that our objective is to obtain features with less correlated noise. Although $k$-hop structural noise still contains the original feature noise, its effect is attenuated, as a consequence of the following result (see Appendix B).

**Proposition 2.** *Given the feature matrix $\mathbf{X}$, let $\overline{\mathbf{X}}$ be the mean feature matrix, where the $i$-th row is $\mathbf{x}_c$ if $v_i$ has label $c$. If the graph $\mathcal{G}$ is sufficiently dense, then as $k$ increases, the features $\widetilde{\mathbf{A}}_{\mathcal{G}}^k\mathbf{X}$ is close to $\widetilde{\mathbf{A}}_{\mathcal{G}}^k\overline{\mathbf{X}}$, with high probability.*

Intuitively, as $\overline{\mathbf{X}}$ is *unambiguous* in the sense that it trivially separates all classes, the "uncertainty" or "noise" of $\widetilde{\mathbf{A}}_{\mathcal{G}}^k\overline{\mathbf{X}}$ is solely from different neighborhood structures of distinct nodes. Therefore,

approximately, the "noise" of $\widetilde{\mathbf{A}}_{\mathcal{G}}^k \mathbf{X}$ is also due to distinct neighborhood structures. The result suggests that feature noise $\mathbf{n}_i$ and structural noise $\mathbf{n}_i^{(k)}, k > 0$ are indeed of different characteristics. In other words, the operator $\widetilde{\mathbf{A}}_{\mathcal{G}}^k$ effectively "replaces" feature noise with structural noise. This decoupling between feature and structural noise becomes more pronounced when the graph construction depends only weakly on the initial node features.

We may verify the above numerically as follows, even for a relatively small $k = 2$. For a node $v_i$, we compute its feature and 2-hop structural noise $\mathbf{n}_i$ and $\mathbf{n}_i^{(2)}$. We reshuffle the index so that the components of $\mathbf{n}_i$ are ordered increasingly, while the same indexing is applied to $\mathbf{n}_i^{(2)}$. Sample examples are shown in Fig. 2. We see that $\mathbf{n}_i^{(2)}$ displays a more random behavior with the given feature index ordering.

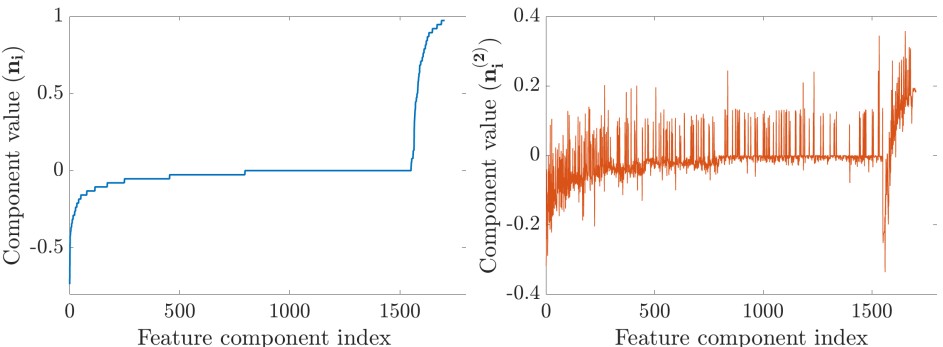

Figure 2: For a sample node $v_i$, the noise $\mathbf{n}_i$ (left panel) and $\mathbf{n}_i^{(2)}$ (right panel) of a random selected node $v_i$ from the Cornell dataset. We see that $\mathbf{n}_i^{(2)}$ displays a more random behavior.

From the examples in Fig. 2, we expect that the correlation between $\mathbf{n}_i^{(2)}$ and $\mathbf{n}_i$ can be reduced due to cancellation from the random spikes. This is desirable, as discussed in Section 2.

Consider the empirical average correlation $E_k = \sum_{1 \leq i \leq N} \langle \mathbf{n}_i, \mathbf{n}_i^{(k)} \rangle / N$ between $\{\mathbf{n}_i\}$ and $\{\mathbf{n}_i^{(k)}\}$, then we have the following observation.

**Observation 1.** *$E_k$ can be decomposed as $E_k = D_k + H_k$, where $D_k$ can be reduced as $k$ increases, while the remaining term $H_k$ has zero expectation, i.e., $\mathbb{E}[H_k] = 0$.*

We emphasize that this statement is heuristic rather than rigorous. A fuller and more explicit explanation is provided in Appendix B. For example, we show rigorously (in Corollary 1) that $D_k$ for $k = 2l + 2$ is always reduced from that for $k = 2l$. This is particularly relevant as 2-layer GCN is commonly used in the GNN literature. The result (cf. Theorem 1) in Appendix B on $E_k$ also further confirms that $\widetilde{\mathbf{A}}_{\mathcal{G}}^k$ transforms "feature noise" into "structural noise".

To summarize, recall we want to comply with the strategies outlined in Section 2. Hence, to generate a "secondary view" to supplement the initial features, it suffices to consider $\widetilde{\mathbf{A}}_{\mathcal{G}}^k$-transformed features. To further enhance the expressiveness, the discussions suggest that we may consider a *simple MLP* and a *simple GCN* (Kipf & Welling, 2017) as the view generation encoders. As a preview, the parameter $k$ corresponds to the number of GCN layers. A moderate choice such as $k = 2$, which conforms to common practices, is usually sufficient to generate less correlated structural noise (see evidence in Fig. 2). We provide more details on the model in the next section.

## 4    VERY SIMPLE GCL

We now present the full details of the proposed simple GCL model. Building on the analysis in the previous sections, the strategy is to use a $k$-layer *GCN* and an *MLP* as view-generation encoders. Consider a graph $\mathcal{G}$ with adjacency matrix $\mathbf{A}$ and node features $\mathbf{X}$, collectively denoted as $\mathcal{X} = (\mathbf{A}, \mathbf{X})$. The $k$-layer *GCN* captures structural features together with their inherent structural

noise, producing the view

$$\mathbf{H}^{(0)} = \mathbf{X}, \ \ \mathbf{H}^{(\ell+1)} = \sigma\left(\widetilde{\mathbf{A}}_{\mathcal{G}}\mathbf{H}^{(\ell)}\mathbf{W}^{(\ell)}\right), \ell = 0, \ldots, k-1,$$

where $\widetilde{\mathbf{A}}_{\mathcal{G}}$ is the normalized adjacency matrix with self-loops, $\mathbf{W}^{(\ell)}$ are learnable weight matrices, and $\sigma(\cdot)$ is a nonlinear activation (e.g., ReLU). The output of the GCN after $k$ layers,

$$\mathbf{Z}_s = \mathbf{H}^{(k)}$$

is a representation with prominent structural noise. In parallel, the *MLP* serves as an encoder that isolates feature noise, generating the feature-noise representation

$$\mathbf{Z}_f = \mathrm{MLP}(\mathbf{X}).$$

Together, $\mathbf{Z}_s$ and $\mathbf{Z}_f$ form two complementary views used for contrastive learning. The learnable parameters of the model are the weight matrices of the GCN and the MLP, while the number of GCN layers $k$ and MLP layers $L$ are treated as hyperparameters. In practice, we adopt $L = 1$ for the MLP, which is a simple and efficient choice and aligns with common practice. To optimize these parameters, we adopt the standard cosmean contrastive loss $\mathcal{L}$ (Thakoor et al., 2022) between $\mathbf{Z}_s$ and $\mathbf{Z}_f$, i.e.,

$$\mathcal{L}(\mathbf{Z}_s, \mathbf{Z}_f) = 1 - \frac{1}{N}\sum_{i=1}^{N}\frac{\langle\mathbf{Z}_{s,i}, \mathbf{Z}_{f,i}\rangle}{\|\mathbf{Z}_{s,i}\|_2\|\mathbf{Z}_{f,i}\|_2},$$

where $\langle\mathbf{Z}_{s,i}, \mathbf{Z}_{f,i}\rangle$ is the inner product of these two vectors $\mathbf{Z}_{s,i}$ and $\mathbf{Z}_{f,i}$, and $\|\mathbf{Z}_{s,i}\|_2$ and $\|\mathbf{Z}_{f,i}\|_2$ are their respective $\ell_2$-norms. It is deemed to align the feature vectors for "amplifying class centroids" (see the first strategy in Section 2).

For downstream tasks, we compute a weighted average of the two views as $\mathbf{Z} = \beta\mathbf{Z}_s + (1 - \beta)\mathbf{Z}_f$, where $\beta$ is either set to 0.5 or tuned based on validation accuracy. This aggregation is effective only if the noise components of $\mathbf{Z}_s$ and $\mathbf{Z}_f$ are not strongly correlated. As in Observation 1, the structure-noise view $\mathbf{Z}_s$ and the feature-noise view $\mathbf{Z}_f$ are less correlated. Consequently, their combination allows the signal components to be reinforced while their independent noise components are partially canceled out. This highlights the importance of constructing diverse feature views that capture different sources of noise, as feature noise and structural noise exhibit inherently different characteristics.

**Heterophily v.s. homophily** While the proposed GCN-MLP model is applicable to both homophilic and heterophilic graphs, its (accuracy) advantage is expected to be more pronounced in the heterophilic setting (cf. Observation 1). In homophilic graphs, conventional GCNs already perform well: since neighbors often share the same label, feature aggregation amplifies class-consistent signals and naturally suppresses noise. In this case, structural and feature information are highly aligned, so the benefit of integrating the two views is less pronounced. However, homophily is a local notion. Even in homophilic datasets, there exists "heterophilic nodes", which are along class boundaries and hence difficult to classify. To evaluate whether our GCN-MLP model provides the additional benefits on such challenging cases, we focus the evaluation on nodes with a heterophily ratio of 1, i.e., nodes whose neighbors all belong to different classes. We report test accuracy on this subset. As shown Table 1, our GCN-MLP consistently outperforms GraphACL, a strong contrastive baseline widely recognized for its performance on homophilic datasets. In addition, in the homophilic setting, the proposed model offers substantial gains in computation and memory efficiency (see Section 5.5), and it further demonstrates strong robustness (see Section 5.4).

Table 1: Test accuracy on heterophilic nodes

|  | Cora | Citeseer | Pubmed |
|---|---|---|---|
| GraphACL | 36.31±0.01 | 31.06±0.87 | 43.43±0.03 |
| GCN-MLP | 39.11±0.65 | 32.95±0.13 | 54.07±0.57 |

In contrast, in heterophilic graphs, neighbors may belong to different classes, and aggregation via message passing amplifies structural noise while suppressing feature noise. By decoupling feature noise and structural noise, the GCN-MLP model produces complementary views: the MLP focuses on extracting information directly from node features, while the GCN leverages the graph structure

to provide a complementary, structurally informed view. Since these two noise sources are less correlated in heterophilic graphs, their combination strengthens useful signals while (partially) canceling independent noise, enabling the model to outperform state-of-the-art GCL methods on challenging heterophilic benchmarks. Numerical evidence supporting this claim is provided in Section 5.2.

**Robustness** In practical scenarios, graphs often exhibit noisy features or incomplete topology (e.g., missing edges) (Lee et al., 2024). The proposed GCN-MLP model mitigates potential performance degradation in such cases. The MLP produces a feature-based view independent of graph topology, so perturbations or missing edges do not affect it. Meanwhile, the structurally informed GCN view is combined with the MLP view under a contrastive objective, which reinforces useful representations while canceling uncorrelated noise. This complementary design enhances robustness to both structural perturbations and feature noise. Empirical results under adversarial attacks in Section 5.4 further demonstrate the model's resilience to both black-box and white-box perturbations.

## 5 EXPERIMENTS

### 5.1 EXPERIMENTAL SETUP

**Datasets and splits** We first focus on heterophilic datasets to verify our claim that our method is particularly well-suited for handling weak or negative homophily, making it more applicable to such scenarios. The benchmarks include Wisconsin, Cornell, Texas, Actor, Crocodile, Squirrel-filtered (Squirrel), and Chameleon-filtered (Chameleon), where the filtered Squirrel and Chameleon versions remove duplicate nodes to avoid training–test leakage (Platonov et al., 2023b). We further evaluate on three large-scale heterophilic datasets: Amazon-ratings (Amazon), Roman-empire (Roman), and Arxiv-year, to test scalability. For the second part, we also report results on homophilic datasets, including citation graphs (Cora, Citeseer, Pubmed) and co-purchase networks (Computer, Photo). All datasets follow the standard public splits, with detailed descriptions in Appendix C.1.

**Baselines** We compare GCN-MLP with a large number of unsupervised learning methods (15 in total), including classical models and recent SOTAs: DGI (Velickovic et al., 2019), GMI (Peng et al., 2020), MVGRL (Hassani & Khasahmadi, 2020), GRACE (Zhu et al., 2020), CCA-SSG (Zhang et al., 2021), BGRL (Thakoor et al., 2022), AFGRL (Lee et al., 2022), DSSL (Xiao et al., 2022), SP-GCL (Wang et al., 2023), GraphACL (Xiao et al., 2023), GraphECL (Xiao et al., 2024), PolyGCL (Chen et al., 2024), LOHA (Zou et al., 2025), EPAGCL (Xu et al., 2025) and SDMG (Zhu et al., 2025). Detailed descriptions and implementations of these baselines are given in Appendix C.2.

**Evaluation protocol** To evaluate the quality of the representation, we mainly focus on the node classification task. Following the standard linear evaluation protocol, we train a linear classifier on the frozen representations and report the test accuracy as the evaluation metric. We further assess GCN-MLP on the graph classification task (see Appendix D.3) to demonstrate its generalization beyond node-level settings.

**Setup** We randomly initialize model parameters and train the encoder with the Adam optimizer. Each experiment is repeated with ten random seeds, and we report the mean performance and standard deviation. For all methods, hyperparameters (i.e., learning rate, weight decay, and hidden feature dimension) are tuned based solely on validation accuracy to ensure fairness, following the settings commonly adopted in standard baselines (Xiao et al., 2023). When baseline results are unavailable for certain datasets or do not follow standard public splits (Xiao et al., 2022; Chen et al., 2024; Zou et al., 2025; Zhu et al., 2025), we reproduce them using the authors' official code.

### 5.2 OVERALL PERFORMANCE

Node classification results on heterophilic and homophilic datasets are reported in Tables 2 and 3, respectively. On heterophilic graphs, GCN-MLP achieves clear state-of-the-art performance, surpassing GraphACL, PolyGCL, GraphECL, and all other baselines by a significant margin. This demonstrates the effectiveness of the strategy that mitigates feature noise via aggregating with representations dominated by weakly correlated structural noise. We refer readers to Appendix D.1 for visualizations and discussions on the relations between model performance and noise correlation.

On the other hand, on homophilic graphs, GCN-MLP provides less pronounced gain in terms of classification accuracy as we have discussed in Section 4. While GCN-MLP shows weaker results

on Cora, it performs comparably to other methods on Citeseer, Pubmed, and Computers, and is on par with the best method (i.e., SDMG) on Photo. Despite a smaller accuracy gain in the homophilic settings, GCN-MLP remains cost-effective as it offers substantial advantages in efficiency, requiring far less computation time and memory, as shown in Table 8 below.

Table 2: Node classification results(%) on heterophilic datasets. The best and the second-best result under each dataset are highlighted in **red** and **blue**, respectively.

| Method | DGI | CCA-SSG | BGRL | DSSL | SP-GCL | GraphACL | PolyGCL | GraphECL | LOHA | EPAGCL | SDMG | GCN-MLP |
|---|---|---|---|---|---|---|---|---|---|---|---|---|
| Squirrel | 40.60±0.35 | 41.23±1.77 | **42.55±2.35** | 40.95±3.35 | 40.11±2.20 | 35.51±2.03 | 33.07±0.94 | 41.14±6.71 | 34.46±1.69 | 40.28±1.59 | 41.55±6.71 | **43.89±1.62** |
| Chameleon | 42.57±0.71 | 39.46±3.10 | 40.13±2.16 | 37.69±2.07 | 44.49±2.59 | 38.59±2.81 | 41.79±2.45 | 35.82±2.76 | **45.45±1.83** | 35.43±1.28 | 36.82±0.77 | **46.01±4.23** |
| Crocodile | 51.25±0.51 | 56.77±0.39 | 53.87±0.65 | 62.98±0.51 | 61.72±0.21 | **66.17±0.24** | 65.95±0.59 | 52.52±3.01 | 66.09±0.69 | 70.14±0.62 | 65.38±0.37 | **66.47±1.20** |
| Actor | 28.30±0.76 | 27.82±0.60 | 28.80±0.54 | 28.15±0.31 | 28.94±0.69 | 30.03±0.13 | 34.37±0.69 | **35.80±0.89** | 33.69±0.73 | 30.02±0.91 | 26.74±0.13 | **36.79±0.91** |
| Wisconsin | 55.21±1.02 | 58.46±0.96 | 51.23±1.17 | 62.25±0.55 | 60.12±0.39 | 69.22±0.40 | 76.08±3.33 | **79.41±2.19** | 77.05±6.08 | 63.73±3.95 | 52.68±1.21 | **85.29±2.19** |
| Cornell | 45.33±6.11 | 52.17±1.04 | 50.33±2.29 | 53.15±1.28 | 52.29±1.21 | 59.33±1.48 | 43.78±3.51 | **69.19±6.86** | 54.05±7.05 | 52.97±5.82 | 45.59±0.67 | **71.35±6.19** |
| Texas | 58.53±2.98 | 59.89±0.78 | 52.77±1.98 | 62.11±1.53 | 59.81±1.33 | 71.08±0.34 | 72.16±3.51 | **75.95±5.33** | 69.73±6.26 | 68.92±5.95 | 53.60±2.67 | **78.38±4.68** |
| Roman | 63.71±0.63 | 67.35±0.61 | 68.66±0.39 | 71.70±0.54 | 70.88±0.35 | **74.91±0.28** | 72.97±0.25 | 45.05±1.57 | OOM | 47.11±0.87 | 49.20±0.51 | **78.21±0.39** |
| Amazon | 42.72±0.42 | 41.23±0.25 | 41.17±0.25 | 42.12±0.78 | 42.04±0.68 | OOM | 44.29±0.43 | 36.88±1.25 | 38.45±0.20 | OOM | **45.18±0.16** | **45.42±0.47** |
| Arxiv-year | 39.26±0.72 | 37.38±0.41 | 43.02±0.62 | 45.80±0.57 | 44.11±0.35 | **47.21±0.39** | 43.07±0.23 | OOM | OOM | OOM | OOM | **46.15±0.08** |

Table 3: Node classification results(%) on homophilic datasets.

| Method | DGI | GMI | MVGRL | GRACE | CCA-SSG | BGRL | AFGRL | SP-GCL | GraphACL | PolyGCL | LOHA | EPAGCL | SDMG | GCN-MLP |
|---|---|---|---|---|---|---|---|---|---|---|---|---|---|---|
| Cora | 82.30±0.60 | 82.70±0.20 | 82.90±0.71 | 80.00±0.41 | **84.00±0.40** | 82.70±0.60 | 82.31±0.42 | 83.16±0.13 | **84.20±0.31** | 82.74±0.14 | 81.22±0.17 | 82.14±0.89 | 83.60±0.60 | 77.26±0.14 |
| Citeseer | 71.80±0.70 | 73.01±0.30 | 72.61±0.70 | 71.72±0.62 | 73.10±0.30 | 71.10±0.80 | 68.70±0.31 | 71.96±0.42 | **73.63±0.22** | 71.82±0.45 | 71.89±0.63 | 71.94±0.57 | **73.20±0.50** | 70.12±0.44 |
| Pubmed | 76.80±0.60 | 80.11±0.22 | 79.41±0.31 | 79.51±1.10 | 81.00±0.40 | 79.60±0.50 | 79.71±0.21 | 79.16±0.84 | **82.02±0.15** | 77.31±0.27 | 78.09±0.29 | **81.28±0.62** | 80.00±0.40 | 79.00±0.03 |
| Computer | 83.95±0.47 | 82.21±0.34 | 87.52±0.11 | 86.51±0.32 | 88.74±0.28 | 89.69±0.37 | **89.90±0.31** | 89.68±0.19 | 89.80±0.25 | 86.54±0.45 | 79.05±0.32 | 76.81±0.79 | **90.40±0.20** | 87.65±1.10 |
| Photo | 91.61±0.22 | 90.72±0.21 | 91.72±0.10 | 92.50±0.22 | 93.14±0.14 | 92.90±0.30 | 93.25±0.33 | 92.49±0.31 | 93.31±0.19 | 91.45±0.35 | 86.46±0.41 | 93.05±0.23 | **94.10±0.20** | **93.41±0.88** |

## 5.3 ABLATION AND HYPERPARAMETER ANALYSIS

To illustrate the complementary roles of the two encoders, the GCN captures structural features with their inherent noise, while the MLP isolates feature noise, together producing complementary views. An ablation study on the Cora, Chameleon, Roman, and Arxiv-year datasets (Table 4) shows that GCN-GCN and MLP-MLP both underperform relative to GCN-MLP, confirming the effectiveness of combining structurally informed and feature-based views. For the selection of key hyperparameters, we largely follow existing literature or the common practice of the GNN/GCL community.

Table 4: Node classification results (%) across different datasets and design configurations.

| Method | Cora | Chameleon | Roman | Arxiv-year |
|---|---|---|---|---|
| MLP-MLP | 64.37±0.31 | 42.13±4.52 | 65.55±0.48 | 35.64±0.28 |
| GCN-GCN | 56.23±0.54 | 38.49±2.72 | 32.83±0.28 | 40.82±0.18 |
| GCN-MLP | 77.26±0.14 | 46.01±4.42 | 77.13±0.46 | 46.15±0.08 |

**Feature dimension** In the GCN-MLP model, the linear layers in both the GCN and MLP expand the feature dimension, thereby increasing representation capacity and enabling the model to capture more complex patterns, as discussed in Xiao et al. (2023). As shown in Fig. 3a, enlarging the feature dimension consistently improves performance on both homophilic and heterophilic graphs, with especially pronounced gains on the latter.

**The number of GCN layers** In the GCN-MLP model, we study the effect of the number of GCN layers $k$ on both homophilic and heterophilic graphs, as shown in Fig. 3b. On heterophilic graphs, performance may improve as $k$ increases, since deeper propagation helps reduce the cosine similarity between features and structural noise (as in Observation 1), thereby enhancing the benefit of combining the two views. In contrast, on homophilic graphs (e.g., Computer), increasing $k$ aggregates class-consistent signals and suppresses noise, so structural and feature information are already strongly correlated. As a result, the advantage of combining the two views becomes limited. Notably, most of the performance gain on heterophilic graphs occurs from $k = 1$ to $k = 2$, while further increases yield diminishing returns. This suggests that a moderate choice such as $k = 2$, consistent with common practice (Kipf & Welling, 2017), is sufficient to reduce structural noise and achieve strong performance without increasing model complexity.

**Augmentation & Negative sampling techniques** To access whether GCN-MLP benefits from additional training tricks, we further evaluate it with two common operations: data augmentation (e.g., edge removal and node-feature masking) and negative sampling using the InfoNCE loss adopted in GRACE, on both homophilic and heterophilic datasets. Table 5 shows that these techniques yield performance comparable to our original model, indicating that GCN-MLP already operates close to its optimal capacity without relying on either augmentation or negative sampling. This aligns with the

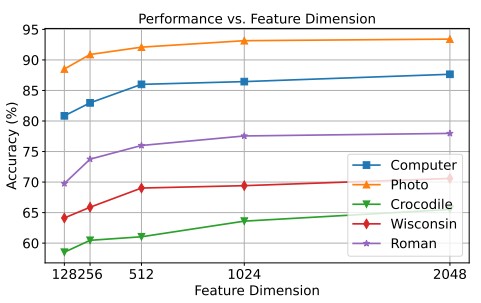 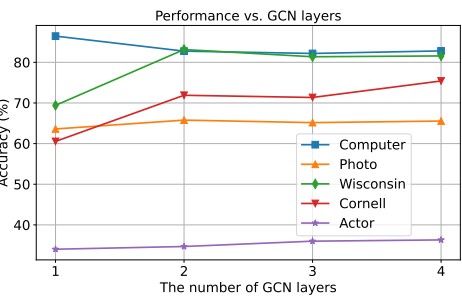

(a) Accuracy vs feature dimension (b) Accuracy vs GCN layers

Figure 3: Performance comparison across feature dimension and GCN layers

principle of Occam's Razor, which favors simpler models when additional complexity does not offer clear benefits. Our augmentation-free and negative-sample-free design therefore remains lightweight, effective, and efficient while maintaining strong performance.

Table 5: Comparison GCN-MLP with its additional techniques

|  | Crocodile | Wisconsin | Roman | Pubmed | Photo |
|---|---|---|---|---|---|
| GCN-MLP | 66.47±1.20 | 85.29±2.19 | 78.21±0.39 | 79.00±0.03 | 93.41±0.87 |
| GCN-MLP(+ Aug.) | 65.56±1.01 | 85.49±2.80 | 78.25±0.49 | 79.05±0.22 | 93.48±0.82 |
| GCN-MLP(+ Neg.) | 65.71±0.95 | 84.92±3.26 | 78.16±0.45 | 78.67±0.13 | 93.40±0.96 |
| GCN-MLP(+ Aug. & Neg.) | 65.71±0.55 | 85.69±4.96 | 78.20±0.49 | 78.58±0.04 | 93.46±0.97 |

## 5.4 ROBUSTNESS STUDY

We evaluate the claimed robustness of GCN-MLP through node classification under both black-box and white-box adversarial attacks on standard benchmarks, including homophilic (Photo) and heterophilic (Actor, Wisconsin, Texas) graphs. GCN-MLP is compared against eight baselines: a robust supervised method (FROND (Kang et al., 2024)), three robust GCL methods (GCL-Jac (Xu et al., 2020), Ariel (Feng et al., 2022), Res-GRACE (Lin et al., 2024)), and five state-of-the-art GCL models (GraphACL, PolyGCL, LOHA, EPAGCL, SDMG). Additional results on more datasets are provided in Appendix D.2.

**Attacks methods** We consider four *black-box* topology attacks in the evasion setting: Random, PRBCD (Zügner et al., 2018), Nettack (Geisler et al., 2021), and Metattack (Zügner & Günnemann, 2019). In addition, we evaluate two *white-box* attacks (i.e., PGD (Madry et al., 2018) and PRBCD) that jointly perturb the graph structure and node features. All models are trained on clean graphs, while adversarial perturbations are introduced only at inference. Further implementation details of the attacks are provided in Appendix C.3.

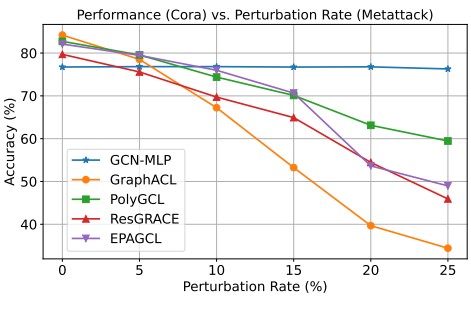 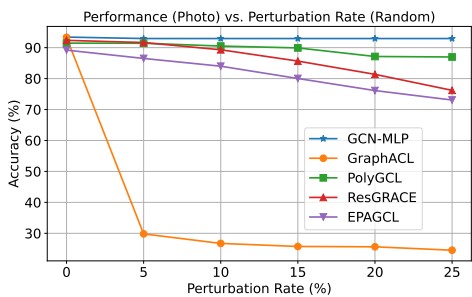

(a) Accuracy vs Metattack rate (b) Accuracy vs Random attack rate

Figure 4: Performance comparison under adversarial attacks on Cora and Photo datasets

**Robust results** We first evaluate GCN-MLP's robustness under increasing perturbation rates (from 0% to 25% in 5% increments), using Metattack on Cora (Fig. 4a) and Random attack on Photo (Fig. 4b). While GCN-MLP performs worse than baselines on Cora in the absence of attacks, it exhibits superior robustness as the perturbation rate increases, consistently outperforming strong baselines (e.g., GraphACL), particularly at higher perturbation levels. This robustness stems from

its dual-view design: the MLP provides a structure-independent feature view resilient to edge perturbations, while the GCN offers a structure-aware view. Contrastive learning aligns the two, reinforcing consistent signals and suppressing adversarial noise, so the feature view anchors stable representations even when the structural view is corrupted. Results in Table 6 and Table 7 further confirm its superior robustness, especially under more challenging white-box attacks.

Table 6: Black-box attack robust accuracy results(%) on graph evasion attack for node classification.

| Dataset | Attack | FROND | GCL-Jac | Ariel | Res-GRACE | GraphACL | PolyGCL | LOHA | EPAGCL | SDMG | GCN-MLP |
|---|---|---|---|---|---|---|---|---|---|---|---|
| Photo | *clean* | 92.93±0.46 | 91.46±0.50 | 85.75±1.21 | 92.23±1.22 | 93.31±0.19 | 91.45±0.35 | 86.46±0.41 | 93.05±0.23 | 94.10±0.20 | 93.41±0.88 |
| | Random | 89.90±1.21 | 86.40±0.74 | 80.62±1.53 | 87.79±1.93 | 26.61±0.05 | 90.17±0.99 | 85.83±1.12 | 84.08±1.50 | 89.90±0.78 | 92.94±0.58 |
| | PRBCD | 88.58±1.05 | 85.24±1.30 | 80.58±1.62 | 85.39±4.19 | 29.13±0.95 | 89.65±0.39 | 86.35±1.07 | 80.60±2.72 | 89.42±0.96 | 92.84±0.28 |
| | Metattack | 89.61±1.13 | 86.20±1.06 | 82.76±1.11 | 85.46±1.56 | 28.42±0.74 | 91.06±1.36 | 86.56±0.89 | 85.65±0.56 | 90.78±0.99 | 91.14±0.68 |
| | Nettack | 91.17±1.35 | 90.50±0.63 | 85.28±0.91 | 91.51±1.40 | 32.84±0.25 | 91.29±1.15 | 87.40±0.89 | 89.59±1.05 | 90.29±0.56 | 92.34±0.52 |
| Wisconsin | *clean* | 67.84±3.84 | 43.53±6.19 | 56.08±4.31 | 52.35±7.18 | 69.22±0.40 | 76.08±3.33 | 76.05±6.08 | 63.73±3.95 | 52.68±1.21 | 85.10±2.35 |
| | Random | 69.61±4.49 | 44.71±6.43 | 51.18±5.44 | 51.76±6.27 | 51.56±5.63 | 75.23±3.13 | 76.47±4.12 | 59.02±4.59 | 51.18±0.98 | 85.29±1.81 |
| | PRBCD | 67.65±5.28 | 44.71±6.72 | 55.88±4.41 | 51.37±6.67 | 52.55±5.13 | 74.60±3.14 | 75.29±4.12 | 60.39±6.61 | 50.98±0.78 | 84.90±2.33 |
| | Metattack | 64.51±5.98 | 43.53±4.09 | 50.98±4.64 | 50.59±6.06 | 52.15±5.08 | 76.67±3.92 | 74.71±4.31 | 60.39±4.79 | 51.67±1.47 | 84.90±1.76 |
| | Nettack | 70.78±6.17 | 44.71±5.32 | 55.29±5.02 | 50.00±5.70 | 53.73±5.16 | 77.65±3.92 | 75.49±3.73 | 59.02±2.97 | 51.57±1.57 | 85.10±2.00 |
| Actor | *clean* | 35.08±1.08 | 29.25±1.21 | 24.36±1.11 | 30.72±0.72 | 30.03±0.13 | 34.37±0.69 | 33.69±0.73 | 30.02±0.91 | 26.74±0.13 | 36.56±0.93 |
| | Random | 35.15±0.78 | 27.59±1.12 | 25.64±1.02 | 30.16±1.09 | 28.36±1.95 | 25.41±0.72 | 34.19±0.59 | 28.92±1.03 | 27.09±0.68 | 36.19±0.77 |
| | PRBCD | 35.04±0.90 | 27.76±1.66 | 24.95±0.89 | 30.48±1.28 | 28.37±1.95 | 27.21±0.64 | 26.23±0.79 | 28.66±2.01 | 26.79±0.82 | 36.47±1.05 |
| | Metattack | 32.34±7.10 | 28.00±1.10 | 25.54±0.75 | 30.34±1.04 | 28.45±1.26 | 28.29±0.42 | 26.97±0.65 | 29.65±1.12 | 26.78±0.91 | 36.56±1.12 |
| | Nettack | 34.97±0.88 | 28.87±0.73 | 25.51±0.95 | 30.86±0.96 | 28.60±1.20 | 25.96±0.86 | 27.20±0.74 | 30.05±0.81 | 26.72±0.79 | 36.14±0.67 |

Table 7: White-box attack robust accuracy (%) under 15% perturbation for node classification.

| Dataset | Attack | FROND | Res-GRACE | GraphACL | PolyGCL | LOHA | EPAGCL | SDMG | GCN-MLP |
|---|---|---|---|---|---|---|---|---|---|
| Photo | *clean* | 92.03±1.27 | 92.50±0.17 | 93.31±0.19 | 91.45±0.35 | 86.46±0.41 | 93.05±0.23 | 94.10±0.20 | 93.41±0.88 |
| | PGD | 14.18±5.16 | 45.46±2.05 | 30.94±3.62 | 22.93±2.73 | 59.35±0.43 | 7.71±1.41 | 3.11±1.03 | 90.56±0.54 |
| | PRBCD | 15.08±7.52 | 40.72±3.70 | 28.56±1.16 | 15.86±1.99 | 52.82±3.45 | 3.69±1.35 | 21.35±11.31 | 75.42±0.69 |
| Texas | *clean* | 74.86±3.21 | 54.59±5.51 | 71.08±0.34 | 72.43±4.86 | 69.73±6.26 | 68.92±4.05 | 53.60±2.67 | 78.38±4.68 |
| | PGD | 69.46±7.16 | 28.85±8.55 | 21.62±7.20 | 56.76±16.22 | 32.70±20.17 | 48.65±17.72 | 27.57±22.67 | 75.68±6.28 |
| | PRBCD | 65.14±3.91 | 26.49±11.22 | 23.78±8.99 | 53.51±13.23 | 37.57±15.17 | 46.22±16.24 | 15.68±16.12 | 67.03±5.10 |
| Wisconsin | *clean* | 67.84±3.84 | 52.35±7.18 | 69.22±0.40 | 76.08±3.33 | 76.05±6.08 | 63.73±3.95 | 52.68±1.21 | 85.10±2.35 |
| | PGD | 62.16±6.01 | 30.45±6.89 | 8.04±3.75 | 66.27±6.06 | 66.86±7.52 | 18.43±10.57 | 15.29±17.60 | 80.98±4.64 |
| | PRBCD | 57.06±5.71 | 27.56±8.36 | 11.37±3.56 | 60.59±5.65 | 57.06±7.36 | 33.92±11.20 | 18.24±12.15 | 73.53±4.04 |
| Cornell | *clean* | 63.24±9.38 | 51.08±5.19 | 59.33±1.48 | 43.78±3.51 | 54.05±7.05 | 52.97±5.82 | 45.59±0.67 | 73.78±5.68 |
| | PGD | 44.36±8.34 | 30.58±6.13 | 29.46±10.23 | 38.65±6.17 | 48.11±5.10 | 20.00±9.98 | 39.46±11.67 | 53.78±8.92 |
| | PRBCD | 43.81±8.00 | 28.13±5.47 | 37.03±8.74 | 36.76±7.47 | 44.05±7.65 | 20.54±13.20 | 13.24±8.24 | 52.16±9.52 |

## 5.5 COMPLEXITY ANALYSIS

The training time complexity of GCN-MLP consists of three parts: solving the GCN encoder, the MLP encoder, and computing the contrastive loss. For a graph with $N$ nodes and $|\mathcal{E}|$ edges, each GCN layer requires $\mathcal{O}(|\mathcal{E}|h + Nh^2)$ operations, where $h$ is the hidden feature dimension, leading to a total of $\mathcal{O}(k(|\mathcal{E}|h + Nh^2))$ for $k$ layers. The MLP encoder adds $\mathcal{O}(LNh^2)$ for $L$ layers. The contrastive loss further requires pairwise similarity computations $\mathcal{O}(N)$, resulting in a total training complexity of $\mathcal{O}(k(|\mathcal{E}|h + Nh^2) + N)$. Training/inference time and memory comparisons with baselines (e.g., GraphACL, PolyGCL, GraphECL, EPAGCL, and SDMG) are shown in Table 8. GCN-MLP consistently achieves much lower training, inference costs and avoids out-of-memory issues, even on large-scale graphs (e.g., Arxiv-year).

Table 8: Comparison of training time (s), inference time (s), and storage (MiB) across different datasets.

| Method | Training Time (s) | | | | Inference Time (s) | | | | Storage (MiB) | | | |
|---|---|---|---|---|---|---|---|---|---|---|---|---|
| | Cora | Wisconsin | Roman | Arxiv-year | Cora | Wisconsin | Roman | Arxiv-year | Cora | Wisconsin | Roman | Arxiv-year |
| GraphACL | 0.22 | 0.63 | 248.38 | 927.48 | 13.29 | 38.69 | 1215.34 | 44.59 | 326 | 204 | 11999 | 6114 |
| PolyGCL | 0.34 | 0.23 | OOM | OOM | 7.46 | 9.73 | OOM | OOM | 4098 | 894 | OOM | OOM |
| GraphECL | 0.02 | 0.27 | 60.58 | OOM | 0.50 | 0.04 | 0.04 | OOM | 112 | 336 | 2306 | OOM |
| EPAGCL | 0.13 | 0.12 | 0.97 | OOM | 2.81 | 2.63 | 3.14 | OOM | 649 | 329 | 22127 | OOM |
| SDMG | 0.06 | 0.54 | 0.36 | OOM | 1.22 | 0.28 | 0.55 | OOM | 2888 | 824 | 3676 | OOM |
| GCN-MLP | 0.04 | 0.007 | 0.097 | 3.96 | 0.02 | 0.001 | 0.031 | 1.89 | 2396 | 588 | 5850 | 44368 |

## 6 CONCLUSION

In conclusion, we introduce a minimal yet effective framework for graph contrastive learning that avoids the complexity of augmentations, negative sampling, and sophisticated encoders. By constructing complementary views from features and graph structure, the proposed GCN-MLP achieves strong performance with low computational and memory overhead. Theoretical analysis supports its foundation, and extensive experiments across a wide variety of graph datasets, including robustness under adversarial attacks, validate its practicality. These results highlight that simplicity, rather than complexity, can drive effective and efficient graph contrastive learning.

## 7 ETHICS STATEMENT

This research relies solely on publicly available benchmark datasets, used in accordance with their licenses, and does not involve human subjects or personally identifiable information. No private or sensitive data were accessed. The methods are intended for academic research and pose no foreseeable risks of harm or misuse. We also take care to ensure fair evaluation by following standard protocols and reporting reproducible results. The authors affirm full compliance with the ICLR Code of Ethics.

## 8 REPRODUCIBILITY STATEMENT

We have taken several steps to ensure the reproducibility of our work. The main text presents the problem formulation, model design, and theoretical analysis, with complete proofs of all theorems and assumptions provided in the appendix. Detailed descriptions of datasets, preprocessing steps, and evaluation protocols are included in the experimental section and supplementary materials. Hyperparameter settings, training details, and additional ablation studies are also reported in either the Appendix or supplementary materials. To further support replication, the source code is provided in the supplementary material. Collectively, these resources enable independent researchers to reproduce both our theoretical and empirical results.

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

# A RELATED WORKS

## A.1 GRAPH CONTRASTIVE LEARNING WITHOUT AUGMENTATION

Deep Graph Infomax (DGI) (Velickovic et al., 2019) is a seminal framework in graph contrastive learning, which maximizes mutual information (MI) between local node features and a global graph representation. It aggregates node features into a global embedding using a readout function, and employs a discriminator to distinguish positive samples from the original graph against negative samples generated by shuffling node features. This corruption serves as an augmentation, boosting robustness and generalization. Contrastive Multi-view Representation Learning (MVGRL) (Hassani & Khasahmadi, 2020) extends this idea by leveraging multiple graph views generated through different graph diffusion processes. Its discriminator contrasts node-level and graph-level embeddings across views, leading to richer representations. Cross-Scale Contrastive Graph Knowledge Synergy (CGKS) (Zhang et al., 2023) further advances this line by constructing a graph pyramid of coarse-grained views and employing a joint optimization strategy with pairwise contrastive loss to transfer knowledge across scales.

GRACE (Zhu et al., 2020) adopts a different strategy by producing two graph views via edge removal and node feature masking, then maximizing agreement between their node embeddings. It further enhances contrastive learning with both inter-view and intra-view negative pairs. GCA (Zhu et al., 2021) improves upon GRACE by designing adaptive augmentations guided by topological and semantic priors. Moving beyond dual views, ASP (Chen & Kou, 2023) introduces three complementary perspectives, i.e., original, attribute, and global, into a joint contrastive learning framework, strengthening representation quality across these perspectives.

GraphCL (You et al., 2020) systematizes augmentation strategies tailored to graph data. To handle non-homophilous graphs, DSSL (Xiao et al., 2022) and HGRL (Chen et al., 2022) exploit global and high-order information. While HGCL relies on augmentations, DSSL assumes an underlying graph generation process, which may not align with real-world scenarios. Despite these advances, augmentation-based methods have notable limitations: their performance is highly sensitive to the chosen augmentations, with no universally optimal strategy. In addition, they tend to bias the encoder toward low-frequency components, while overlooking high-frequency information that is essential for learning on heterophilic graphs (Liu et al., 2022). More recently, EPAGCL (Xu et al., 2025) combines edge addition and deletion, generating augmented views by adding or dropping edges according to weights derived from the Error Passing Rate (EPR).

To overcome the drawbacks of augmentation-based methods, augmentation-free approaches have been proposed. Graphical Mutual Information (GMI) (Peng et al., 2020) directly estimates MI between input features and representations of nodes and edges, eliminating the need for data augmentation. L-GCL (Zhang et al., 2022) also avoids augmentations but focuses primarily on homophilic graphs. SP-GCL (Wang et al., 2023) overcomes this by capturing both low- and high-frequency components, making it effective for heterophilic structures. GraphACL (Xiao et al., 2023) further removes reliance on both augmentations and homophily assumptions, achieving robust performance across varying graph types. More recent methods adopt spectral strategies to replace augmentation entirely. PolyGCL (Chen et al., 2024) employs learnable polynomial filters to construct spectral views with varying frequency responses. LOHA (Zou et al., 2025) directly contrasts natural low- and high-pass components in the spectral domain to facilitate contrastive learning. Similarly, AFECL (Li et al., 2025) introduces an edge-centric contrastive framework that operates without any form of augmentation. SDMG (Zhu et al., 2025) employs two dedicated low-frequency encoders to extract global signals, promoting a diffusion-based self-supervised learning scheme.

Although SimMLP (Wang et al., 2025) and GraphECL (Xiao et al., 2024) also employ GCN and MLP as dual encoders, our GCN-MLP framework differs fundamentally in both motivation and contrastive formulation. SimMLP and GraphECL distill knowledge from a teacher GNN into a student MLP trained solely on node features, with the aim of integrating rich structural information into the MLP. Only the MLP outputs are used for downstream tasks, with the primary goal of accelerating inference by replacing GNN computation. In contrast, GCN-MLP is not a distillation model but is grounded in a new principle: feature noise and structural noise are weakly correlated, and their contrastive interaction leads to stronger noise cancellation. Therefore, our design goal is to construct two views whose noise components are as uncorrelated as possible. The GCN-MLP architecture is a simple yet effective instantiation of this principle, where the GCN encodes structural information

with its associated structural noise, and the MLP isolates feature noise. This naturally facilitates noise cancellation through contrastive learning and linear combination, resulting in cleaner and more discriminative features for node classification, particularly on challenging heterophilic graphs.

### A.2   GRAPH CONTRASTIVE LEARNING WITHOUT NEGATIVE SAMPLE PAIRS

Building on the success of BYOL for image data, BGRL (Thakoor et al., 2022) eliminates the need for negative samples in graph contrastive learning. It generates two graph augmentations through random node feature masking and edge masking, using an online encoder and a target encoder. The objective is to maximize the cosine similarity between the online encoder's prediction and the target encoder's embedding. To prevent mode collapse and ensure stable training, a stop-gradient operation is applied to the target encoder.

Augmentation-Free Graph Representation Learning (AFGRL) (Lee et al., 2022) addresses the limitations of augmentation-dependent methods like BGRL and GCA (Zhu et al., 2021), where representation quality heavily depends on the choice of augmentation schemes. Building on the BGRL framework, AFGRL eliminates the need for augmentations by generating positive samples directly from the original graph for each node. This approach captures both local structural information and global semantics. However, it introduces higher computational costs.

Inspired by Canonical Correlation Analysis (CCA) methods (Hardoon et al., 2004), CCA-SSG (Zhang et al., 2021) introduces an unsupervised learning framework for graphs without relying on negative sample pairs. It maximizes the correlation between two augmented views of the same input while decorrelating the feature dimensions within a single view's representation.

These advancements highlight promising alternatives to traditional graph contrastive learning methods. Employing augmentation-free frameworks or innovative masking strategies mitigates challenges associated with negative sample selection and augmentation dependency, offering robust solutions for graph representation learning.

## B   DISCUSSIONS AND PROOFS

*Proof of Proposition 1.* We have the following simple observation: let $\mathbf{v}_1, \mathbf{v}_2$ and $0 < \beta < 1$. Then $\|\beta\mathbf{v}_1 + (1 - \beta)\mathbf{v}_2\|$ is a non-decreasing function of the cosine similarity between $\mathbf{v}_1$ and $\mathbf{v}_2$, while keeping their norms fixed. This follows from laws of cosines: $\|\beta\mathbf{v}_1 + (1 - \beta)\mathbf{v}_2\|^2 = \beta^2\|\mathbf{v}_1\|^2 + (1 - \beta)^2\|\mathbf{v}_2\|^2 + 2\beta(1 - \beta)\|\mathbf{v}_1\|\|\mathbf{v}_2\|\cos(\alpha)$, where $\alpha$ is the angle between $\mathbf{v}_1, \mathbf{v}_2$. Therefore, $\|\beta\mathbf{v}_1 + (1 - \beta)\mathbf{v}_2\|$ is non-decreasing in $\cos(\alpha)$.

The first part of Proposition 1 follows from letting $\mathbf{v}_1 = \mathbf{z}_{1,c}$ and $\mathbf{v}_2 = \mathbf{z}_{2,c}$. For the second part of Proposition 1, notice that $\mathbf{n}_1 = \beta\mathbf{n}'_{1,1} + (1-\beta)\mathbf{n}'_{2,1}$. Hence, it suffices to apply the above observation with $\mathbf{v}_1 = \mathbf{n}'_{1,1}$ and $\mathbf{v}_2 = \mathbf{n}'_{2,1}$. $\qquad\square$

*Outline of the proof of Proposition 2.* This result is essentially Ji et al. (2025, Theorem 1(a)). We indicate the underlying reason here, and readers are referred to Ji et al. (2025) for technical details and assumptions. We notice that the operator $\widetilde{\mathbf{A}}_{\mathcal{G}}^k$ is an averaging operator of feature vectors, then one may apply the vector Bernstein inequality (Kohler & Lucchi, 2017, Lemma 18) to obtain the desired noise mitigation. $\qquad\square$

We next discuss Observation 1. From the above proof, we notice that the term $\|\mathbf{v}_1\|\|\mathbf{v}_2\|\cos(\alpha)$ (in the law of cosines) is essentially the inner product $\langle\mathbf{v}_1, \mathbf{v}_2\rangle$, which plays the key role in the analysis of $\|\beta\mathbf{v}_1 + (1 - \beta)\mathbf{v}_2\|^2$. Therefore, for the rest of the appendix, we use $\langle\mathbf{v}_1, \mathbf{v}_2\rangle$ to quantify the correlation between $\mathbf{v}_1$ and $\mathbf{v}_2$.

Recall that the normalized Laplacian matrix $\widetilde{\mathbf{L}}_{\mathcal{G}}$ is defined as $I_N - \widetilde{\mathbf{A}}_{\mathcal{G}}$, where $I_N$ is the identity matrix. It is symmetric and hence admits an orthogonal eigenbasis, i.e., $\widetilde{\mathbf{L}}_{\mathcal{G}} = \mathbf{W}_{\mathcal{G}}\mathbf{\Lambda}_{\mathcal{G}}\mathbf{W}_{\mathcal{G}}^{\mathsf{T}}$, where columns $\mathbf{w}_i, i \leq N$ of $\mathbf{W}_{\mathcal{G}}$ are eigenvectors and their associated eigenvalues are $\lambda_i, 1 \leq N$. They are ordered increasingly and form the diagonal of $\mathbf{\Lambda}_{\mathcal{G}}$.

For the $j$-th feature component, let $\mathbf{m}_j$ be the column vector whose $i$-th entry is the $j$-th component of $\mathbf{n}_i$, the feature noise of node $v_i$. Consider $\widetilde{\mathbf{L}}_{\mathcal{G}}$ as the graph shift operator (Shuman et al., 2013).

Then the $i$-th Fourier coefficient $\widehat{\mathbf{m}}_j(i)$ of $\mathbf{m}_j$ is the number $\langle \mathbf{w}_i, \mathbf{m}_j \rangle$. According to the general principle of graph signal processing, if $\mathbf{m}_j$ is smooth, then $\widehat{\mathbf{m}}_j(i)$ is relatively small for large $i$ and relatively large for small $i$.

**Theorem 1.** *Let the empirical average correlation between the feature noise and structural noise be*

$$E_k = \frac{1}{N} \sum_{1 \le i \le N} \langle \mathbf{n}_i, \mathbf{n}_i^{(k)} \rangle.$$

*There is a decomposition $E_k = D_k + H_k$ such that the following holds:*

*(a)*

$$D_k = \frac{1}{N} \sum_{1 \le i \le N} (1 - \lambda_i)^k \sum_{1 \le j \le N} \widehat{\mathbf{m}}_j(i)^2,$$

*where $\{\lambda_i : i = 1, \ldots, N\}$ are the eigenvalues of the normalized Laplacian $\tilde{\mathbf{L}}_{\mathcal{G}}$ and $\{\widehat{\mathbf{m}}_j(i) : i, j = 1, \ldots, N\}$ are the Fourier coefficients of the feature noise matrix.*

*(b) The term $H_k$ takes the form*

$$H_k = \frac{1}{N} \sum_{1 \le i \le N} \langle \mathbf{n}_i, \mathbf{g}_i \rangle,$$

*such that $\mathbf{g}_i$ depends only on the graph topology and class centroids. Furthermore, $\mathbb{E}[H_k] = 0$.*

*Proof.* Let $\mathbf{N}$ be the matrix whose $i$-th row is $\mathbf{n}_i$, denoted by $(\mathbf{N})_i$. Recall that $\mathbf{n}_i = \mathbf{x}_i - \mathbf{x}_c$, where $c$ is the class label of $v_i$. We re-express $\mathbf{n}_i^{(k)}$ in (1) as

$$\mathbf{n}_i^{(k)} = (\widetilde{\mathbf{A}}_{\mathcal{G}}^k \mathbf{X})_i - (\widetilde{\mathbf{A}}_{\mathcal{G}}^k \overline{\mathbf{X}})_i + (\widetilde{\mathbf{A}}_{\mathcal{G}}^k \overline{\mathbf{X}})_i - \mathbf{M}_i$$
$$= (\widetilde{\mathbf{A}}_{\mathcal{G}}^k \mathbf{N})_i + \mathbf{g}_i, \tag{3}$$

where $\mathbf{g}_i$ is the $i$-th row of $\widetilde{\mathbf{A}}_{\mathcal{G}}^k \overline{\mathbf{X}} - \mathbf{M}$. Therefore, we have

$$E_k = \frac{1}{N} \sum_{1 \le i \le N} \langle \mathbf{n}_i, (\widetilde{\mathbf{A}}_{\mathcal{G}}^k \mathbf{N})_i \rangle + \frac{1}{N} \sum_{1 \le i \le N} \langle \mathbf{n}_i, \mathbf{g}_i \rangle.$$

Therefore, we have $E_k = D_k + H_k$, with the following respective expressions:

$$D_k = \frac{1}{N} \sum_{1 \le i \le N} \langle \mathbf{n}_i, (\widetilde{\mathbf{A}}_{\mathcal{G}}^k \mathbf{N})_i \rangle, \text{ and } H_k = \frac{1}{N} \sum_{1 \le i \le N} \langle \mathbf{n}_i, \mathbf{g}_i \rangle.$$

It suffices to show that they have the stated properties in (a) and (b).

For (a), we may express

$$N D_k = \text{Tr}(\mathbf{N}(\widetilde{\mathbf{A}}_{\mathcal{G}}^k \mathbf{N})^\mathsf{T}) = \text{Tr}((\widetilde{\mathbf{A}}_{\mathcal{G}}^k \mathbf{N})^\mathsf{T} \mathbf{N}).$$

Notice that $\widetilde{\mathbf{A}}_{\mathcal{G}} = I_N - \widetilde{\mathbf{L}}_{\mathcal{G}}$ has the same eigenvectors as $\widetilde{\mathbf{L}}_{\mathcal{G}}$, while the eigenvalues are of the form $1 - \lambda_i$. Let $\mathbf{\Gamma}_{\mathcal{G}}$ be the diagonal matrix whose diagonal entries are $1 - \lambda_i, i \le N$. Then we have the following:

$$N D_k = \text{Tr}((\mathbf{W}_{\mathcal{G}} \mathbf{\Gamma}_{\mathcal{G}}^k \mathbf{W}_{\mathcal{G}}^\mathsf{T} \mathbf{N})^\mathsf{T} \mathbf{N}) = \text{Tr}((\mathbf{W}_{\mathcal{G}}^\mathsf{T} \mathbf{N})^\mathsf{T} \mathbf{\Gamma}_{\mathcal{G}}^k (\mathbf{W}_{\mathcal{G}}^\mathsf{T} \mathbf{N})).$$

Notice that the $(i, j)$-th entry of $\mathbf{W}_{\mathcal{G}}^\mathsf{T} \mathbf{N}$ is the Fourier coefficient $\widehat{\mathbf{m}}_j(i)$. Therefore the $i$-th diagonal entry of $(\mathbf{W}_{\mathcal{G}}^\mathsf{T} \mathbf{N})^\mathsf{T} \mathbf{\Gamma}_{\mathcal{G}}^k (\mathbf{W}_{\mathcal{G}}^\mathsf{T} \mathbf{N})$ is $(1 - \lambda_i)^k \sum_{1 \le k \le N} \widehat{\mathbf{m}}_j(i)^2$. Therefore, we have:

$$N D_k = \text{Tr}((\mathbf{W}_{\mathcal{G}}^\mathsf{T} \mathbf{N})^\mathsf{T} \mathbf{\Gamma}_{\mathcal{G}}^k (\mathbf{W}_{\mathcal{G}}^\mathsf{T} \mathbf{N})) = \sum_{1 \le i \le N} (1 - \lambda_i)^k \sum_{1 \le k \le N} \widehat{\mathbf{m}}_j(i)^2.$$

This proves the claim for (a).

To show (b), we note that $\mathbb{E}[\langle \mathbf{n}_i, \mathbf{g}_i \rangle] = \langle \mathbb{E}[\mathbf{n}_i], \mathbf{g}_i \rangle = 0$ since $\mathbf{g}_i$ is deterministic from (3) and $\mathbb{E}[\mathbf{n}_i] = 0$. Therefore, $\mathbb{E}[H_k] = 0$.

$\square$

We can say more about the summand $D_k$. The eigenvalues $\lambda_i, 1 \le i \le N$ are known to belong to $[0, 2]$. Hence, the following holds:

**Corollary 1.** *For all $l \ge 0$, the sequence $D_{2l}$ is monotonically decreasing, i.e., $D_{2l+2} \le D_{2l}$.*

For general $k$, the trend depends on the size $\widehat{\mathbf{m}}_j(i)$ for different $i$. In particular, if the signal is more concentrated on the low-frequency components, i.e., $1 - \lambda_i \ge 0$, then an average reduction in $D_k$ should be observed for general $k$. In the homophilic setting, due to smoothness, the signal is likely concentrated for those frequency components where $1 - \lambda_i \approx 1$. Therefore, the reduction in $D_k$ is expected to be less pronounced.

If the summand $D_k$ is (made) small, then the average correlation $E_k$ is dominated by $H_k$. Since $\mathbf{g}_i$ depends only on the graph structure and class centroids, it is deterministic once the graph and labels are fixed; therefore, we obtain $\mathbb{E}[H_k] = 0$. Consequently, when $E_k \approx H_k$, then $\mathbf{n}_i^{(k)}$ is effectively replaced by $\mathbf{g}_i$ in the correlation computation. As $\mathbf{g}_i$ has no expected alignment with the initial feature noise $\mathbf{n}_i$, the two components are decoupled in expectation, resulting in weak correlation between them.

## C EXPERIMENTAL DETAILS

### C.1 DETAILS OF DATASETS

We refer the reader to Table 9 for detailed statistics of the datasets. Detailed descriptions of the datasets are given below:

Table 9: Statistics of heterophilic and homophilic graph datasets

| Dataset | Nodes | Edges | Classes | Node Features | Data splits |
|---|---|---|---|---|---|
| Texas | 183 | 309 | 5 | 1793 | 48%/32%/20% |
| Cornel | 183 | 295 | 5 | 1703 | 48%/32%/20% |
| Wisconsin | 251 | 466 | 5 | 1703 | 48%/32%/20% |
| Squirrel-filtered | 2205 | 46557 | 5 | 2089 | 48%/32%/20% |
| Chameleon-filtered | 864 | 7754 | 5 | 2325 | 48%/32%/20% |
| Actor | 7600 | 33391 | 5 | 932 | 48%/32%/20% |
| Roman-empire | 22662 | 32927 | 18 | 300 | 50%/25%/25% |
| Amazon-ratings | 24492 | 186100 | 5 | 300 | 50%/25%/25% |
| Arxiv-year | 169343 | 1166243 | 5 | 128 | 50%/25%/25% |
| Cora | 2708 | 5429 | 7 | 1433 | standard |
| Citeseer | 3327 | 4732 | 6 | 3703 | standard |
| PubMed | 19717 | 88651 | 3 | 500 | standard |
| Computer | 13752 | 574418 | 10 | 767 | 10%/10%/80% |
| Photo | 7650 | 119081 | 8 | 745 | 10%/10%/80% |

**Texas, Wisconsin and Cornell** (Rozemberczki et al., 2021). These datasets are webpage networks collected by Carnegie Mellon University from computer science departments at various universities. In each network, nodes represent web pages, and edges denote hyperlinks between them. Node features are derived from bag-of-words representations of the web pages. The task is to classify nodes into five categories: student, project, course, staff, and faculty.

**Chameleon, Crocodile and Squirrel** (Rozemberczki et al., 2021). These datasets represent Wikipedia networks, with nodes corresponding to web pages and edges denoting hyperlinks between them. Node features are derived from prominent informative nouns on the pages, while node labels reflect the average daily traffic of each web page. The *Squirrel-filtered* and *Chameleon-filtered* variants remove duplicate nodes to prevent training–test leakage (Platonov et al., 2023b).

**Actor** (Pei et al., 2020). This dataset is an actor-induced subgraph extracted from the film-director-actor-writer network. Nodes represent actors, and edges indicate their co-occurrence on the same Wikipedia page. Node features are derived from keywords on the actors' Wikipedia pages, while labels categorize the actors into five groups based on the content of their Wikipedia entries.

For **Texas, Wisconsin, Cornell, Chameleon, Crocodile, Squirrel, and Actor** datasets, we utilize the raw data provided by Geom-GCN (Pei et al., 2020) with the standard fixed 10-fold split

for our experiments. These datasets are available for download at: `https://github.com/graphdml-uiuc-jlu/geom-gcn`.

**Roman-empire** (Platonov et al., 2023a) is a heterophilous graph derived from the English Wikipedia article on the Roman Empire. Each node represents a word (possibly non-unique) in the text, with features based on word embeddings. Node classes correspond to syntactic roles, with the 17 most frequent roles as distinct classes, and all others grouped into an 18th class. Following (Platonov et al., 2023a), we use the fixed 10 random splits with a 50%/25%/25% ratio for training, validation, and testing.

**Arxiv-year** (Lim et al., 2021) is a citation network derived from a subset of the Microsoft Academic Graph, focusing on predicting the publication year of papers. Nodes represent papers, and edges indicate citation relationships. Node features are computed as the average of word embeddings from the titles and abstracts. Following (Lim et al., 2021), the dataset is split into training, validation, and testing sets with a 50%/25%/25% ratio.

**Cora, Citeseer, and Pubmed** (Kipf & Welling, 2017). These datasets are among the most widely used benchmarks for node classification. Each dataset represents a citation graph with high homophily, where nodes correspond to documents and edges represent citation relationships. Node class labels reflect the research field, and node features are derived from a bag-of-words representation of the abstracts. The public dataset split is used for evaluation, with 20 nodes per class designated for training, and 500 and 1,000 nodes fixed for validation and testing, respectively.

**Computer and Photo** (Thakoor et al., 2022; McAuley et al., 2015). These datasets are co-purchase graphs from Amazon, where nodes represent products, and edges connect products frequently bought together. Node features are derived from product reviews, while class labels correspond to product categories. Following the experimental setup in Zhang et al. (2022), the nodes are randomly split into training, validation, and testing sets, with proportions of 10%, 10%, and 80%, respectively.

## C.2 BASELINES

**DGI** (Velickovic et al., 2019): Deep Graph InfoMax (DGI) is an unsupervised learning method that maximizes mutual information between node embeddings and a global graph representation. It employs a readout function to generate the graph-level summary and a discriminator to distinguish between positive (original) and negative (shuffled) node-feature samples, enabling effective graph representation learning.

**GMI** (Peng et al., 2020): Graphical Mutual Information (GMI) measures the mutual information between input graphs and hidden representations by capturing correlations in both node features and graph topology. It extends traditional mutual information computation to the graph domain, ensuring comprehensive representation learning.

**MVGRL** (Hassani & Khasahmadi, 2020): Contrastive Multi-View Representation Learning (MVGRL) leverages multiple graph views generated through graph diffusion processes. It contrasts node-level and graph-level representations across these views using a discriminator, enabling robust multi-view graph representation learning.

**GRACE** (Zhu et al., 2020): Graph contrastive representation learning (GRACE) model generates two correlated graph views by randomly removing edges and masking features. It focuses on contrasting node embeddings across these views using contrastive loss, maximizing their agreement while incorporating inter-view and intra-view negative pairs, without relying on injective readout functions for graph embeddings.

**CCA-SSG** (Zhang et al., 2021): Canonical Correlation Analysis inspired Self-Supervised Learning on Graphs (CCA-SSG) is a graph contrastive learning model that enhances node representations by maximizing the correlation between two augmented views of the same graph while reducing correlations across feature dimensions within each view.

**BGRL** (Thakoor et al., 2022): Bootstrapped Graph Latents (BGRL) is a graph representation learning method that predicts alternative augmentations of the input using simple augmentations, eliminating the need for negative examples.

**AFGRL** (Lee et al., 2022): Augmentation-Free Graph Representation Learning (AFGRL) builds on the BGRL framework, avoiding augmentation schemes by generating positive samples directly from the original graph. This approach captures both local structural and global semantic information, offering an alternative to traditional graph contrastive methods, though at the cost of increased computational complexity.

**DSSL** (Xiao et al., 2022): Decoupled self-supervised learning (DSSL) is a flexible, encoder-agnostic representation learning framework that decouples diverse neighborhood contexts using latent variable modeling, enabling unsupervised learning without requiring augmentations.

**SP-GCL** (Wang et al., 2023): Single-Pass Graph Contrastive Learning (SP-GCL) is a single-pass graph contrastive learning method that leverages the concentration property of node representations, eliminating the need for graph augmentations.

**GraphACL** (Xiao et al., 2023): Graph Asymmetric Contrastive Learning (GraphACL) is a simple and effective graph contrastive learning approach that captures one-hop neighborhood context and two-hop monophily similarities in an asymmetric learning framework, without relying on graph augmentations or homophily assumptions.

**PolyGCL** (Chen et al., 2024): It is a graph contrastive learning pipeline that leverages polynomial filters with learnable parameters to generate low-pass and high-pass spectral views, achieving contrastive learning without relying on complex data augmentations.

**GraphECL** (Xiao et al., 2024): It is a simple and efficient contrastive learning method that eliminates message passing during inference by coupling an MLP with a GNN, enabling the MLP to efficiently mimic the GNN's computations, but this design limits representational flexibility and still relies on negative samples for training.

**LOHA** (Zou et al., 2025): It is a self-supervised graph spectral contrastive framework that directly contrasts low-pass and high-pass views based on their natural distinct specialties without additional data augmentations.

**EPAGCL** (Xu et al., 2025): Error-Passing-based Graph Contrastive Learning (EPAGCL) is an augmentation-based GCL model that generates views by adding or dropping edges according to weights derived from the Error Passing Rate (EPR).

**SDMG** (Zhu et al., 2025): Smooth Diffusion Model for Graphs (SDMG) is a novel self-supervised framework that learns recognition-oriented representations without labels, employing two dedicated low-frequency encoders, one for node features and another for topology, to distill global low-frequency signals.

### C.3  ATTACK METHODS

We consider four *black-box* topology attacks in the evasion setting: Random, PRBCD (Zügner et al., 2018), Nettack (Geisler et al., 2021), and Metattack (Zügner & Günnemann, 2019). Additionally, we further consider two *white-box* attacks (i.e., PGD (Madry et al., 2018) and PRBCD) that jointly perturb both the graph structure and node features. A detailed description of these attack methods is provided below.

**Random attack:**  Adds noisy edges by randomly selecting node pairs across the graph. The number of edges inserted is determined by a perturbation ratio with respect to the original edge count.

**PRBCD:**  The Projected Randomized Block Coordinate Descent (PRBCD) attack perturbs the adjacency matrix $\mathbf{A}$ by iteratively adding or removing edges to maximize the classification loss of a surrogate GNN (e.g., GCN). It employs a projected randomized block coordinate descent strategy with a fixed budget of edge modifications, ensuring efficient and scalable adversarial perturbations. In the white-box setting, PRBCD extends naturally to jointly perturb node features by exploiting full access to model parameters and gradients.

**Nettack:**  A targeted structure-based attack designed to mislead node classification. It manipulates the graph by removing edges to same-class nodes, thereby lowering classification confidence, and

by adding edges to different-class nodes to trigger misclassification. Using a surrogate GNN for guidance, it greedily selects the most impactful edge modifications within a fixed budget.

**Metattack:** A global structure-based attack that perturbs the adjacency matrix $\mathbf{A}$ by leveraging meta-gradients of a surrogate GNN. It modifies the graph to maximize overall classification loss, thereby degrading performance across all nodes.

**PGD:** The Projected Gradient Descent (PGD) attack is a white-box method that jointly perturbs graph structure and node features to maximize the target model's classification loss. It applies iterative gradient-based updates within a fixed perturbation budget, projecting modifications back into the feasible space after each step. With full access to model parameters and gradients, PGD delivers strong and precise attacks.

## D    MORE NUMERICAL RESULTS

### D.1    PERFORMANCE AND NOISE CORRELATION

We illustrate that if the two noise sources, namely feature and structural noise, are less correlated, then the resulting GCN-MLP has a better performance. For this, we empirically verify that aggregating feature representations with weakly correlated structural representations helps mitigate feature noise.

We visualize the cosine similarity histograms between structural features (together with inherent structural noise, captured by the GCN) and node feature noise (isolated by the MLP) on three datasets: Cornell, Roman, and Cora, with $k = 1$ or $k = 2$ GCN layers. The results are shown in Fig. 7, Fig. 5, and Fig. 6, respectively.

In general, we always observe that higher accuracy is associated with weaker correlation. Taking Cornell as an example, when the GCN has $k = 2$ layers, the cosine similarity histogram shifts from being concentrated near 1 (strong correlation) toward 0 (weak correlation), compared with $k = 1$. Performance improves significantly, which agrees with our discussions.

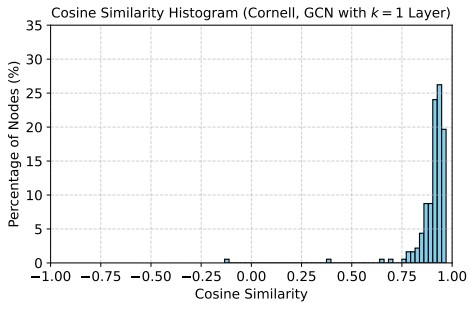 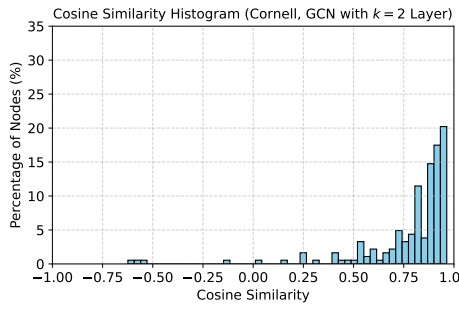

(a) Performance (Cornell): $58.38 \pm 3.86$    (b) Performance (Cornell): $75.14 \pm 3.58$

Figure 5: Performance v.s. noise correlation on Cornell

### D.2    MORE ROBUSTNESS RESULTS

To further assess the robustness of GCN-MLP, we perform node classification under black-box attacks on additional homophilic and heterophilic datasets (e.g., Photo, Citeseer, Wisconsin, Cornell, Texas, and Actor). As shown in Table 10, the results reinforce the robustness of GCN-MLP across a broader range of benchmarks.

### D.3    GRAPH CLASSIFICATION RESULTS

While most GCL methods target node-level representation learning and do not provide a straightforward graph-level extension, we assess GCN-MLP's generality by applying a simple, non-parametric

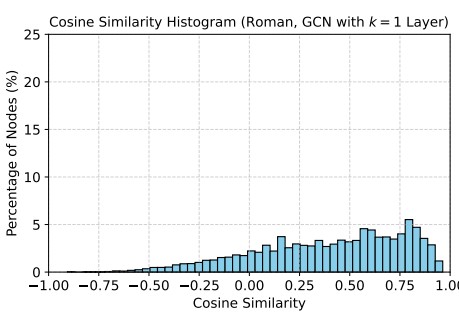

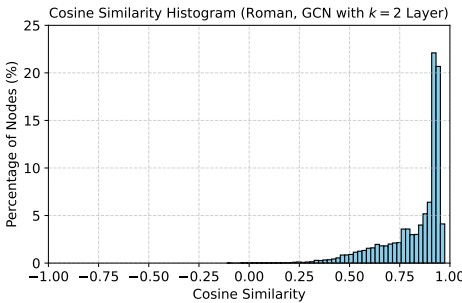

(a) Performance (Roman): $77.74 \pm 0.44$

(b) Performance (Roman): $67.12 \pm 0.51$

Figure 6: Performance v.s. noise correlation on Roman

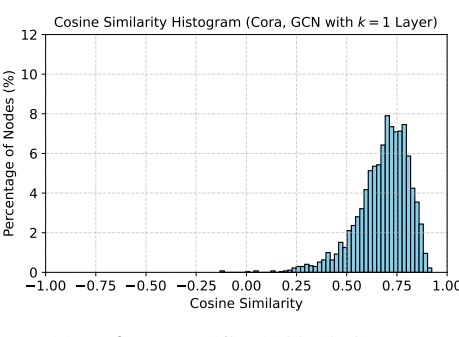

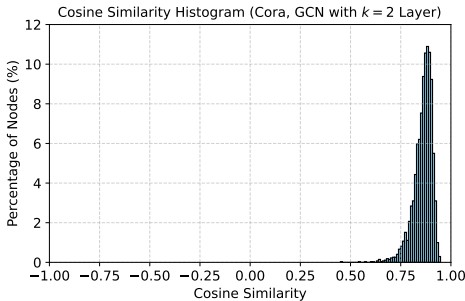

(a) Performance (Cora): $77.37 \pm 0.18$

(b) Performance (Cora): $71.31 \pm 0.30$

Figure 7: Performance v.s. noise correlation on Cora

readout (MeanPooling) to obtain graph-level embeddings. We evaluate this configuration on two standard graph-classification benchmarks, Proteins and DD, and compare against recent graph-level GCL models such as GraphCL and DRGCL as well as strong node-level baselines adapted to the graph-level setting (e.g., GraphACL). GCN-MLP achieves competitive performance compared with node-level baselines and yields results on par with specialized graph-level contrastive methods, demonstrating that our proposed GCN-MLP framework generalizes effectively beyond node-level tasks.

# E    LLM USAGE

We acknowledge the use of large language models (LLMs) as a general-purpose assistive tool in preparing this manuscript. Specifically, LLMs were employed to aid in polishing the writing, including refining grammar, improving clarity, and enhancing fluency of expression. LLMs were **NOT** used for generating research ideas, conducting analysis, or producing results. All conceptual contributions, theoretical developments, experimental designs, and interpretations presented in this work are entirely the responsibility of the authors.

Table 10: Black-box attack robust accuracy results(%) on graph evasion attack for node classification.

| Dataset | Attack | FROND | GCL-Jac | Ariel | Res-GRACE | GraphACL | PolyGCL | LOHA | EPAGCL | SDMG | GCN-MLP |
|---|---|---|---|---|---|---|---|---|---|---|---|
| Photo | *clean* | 92.93±0.46 | 91.46±0.50 | 85.75±1.21 | 92.23±1.22 | 93.31±0.19 | 91.45±0.35 | 86.46±0.41 | 93.05±0.23 | **94.10±0.20** | **93.41±0.88** |
| | Random | 89.90±1.21 | 86.40±0.74 | 80.62±1.53 | 87.79±1.93 | 26.61±0.05 | **90.17±0.99** | 85.83±1.12 | 84.08±1.50 | 89.90±0.78 | **92.94±0.58** |
| | PRBCD | 88.58±1.05 | 85.24±1.30 | 80.58±1.62 | 85.39±4.19 | 29.13±0.95 | **89.65±0.39** | 86.35±1.07 | 80.60±2.72 | 89.42±0.96 | **92.84±0.28** |
| | Metattack | 89.61±1.13 | 86.20±1.06 | 82.76±1.11 | 85.46±1.56 | 28.42±0.74 | **91.06±1.36** | 86.56±0.89 | 85.65±0.56 | 90.78±0.99 | **91.14±0.68** |
| | Nettack | 91.17±1.35 | 90.50±0.63 | 85.28±0.91 | 91.51±1.40 | 32.84±0.25 | **91.29±1.15** | 87.40±0.89 | 89.59±1.05 | 90.29±0.56 | **92.34±0.52** |
| Citeseer | *clean* | 71.37±1.34 | 70.52±0.65 | 50.89±3.76 | 71.72±0.62 | **73.60±0.70** | 71.82±0.45 | 71.95±0.45 | 71.94±0.57 | **73.20±0.50** | 70.12±0.44 |
| | Random | 70.23±1.40 | 57.26±4.20 | 44.98±3.45 | 56.69±2.63 | 68.13±0.44 | **71.58±0.24** | **71.70±0.29** | 63.10±1.08 | 71.47±0.47 | 69.90±0.00 |
| | PRBCD | 71.47±1.29 | 58.30±4.11 | 46.02±3.16 | 58.86±266 | 70.52±1.16 | 71.19±0.61 | **71.60±0.63** | 64.54±2.00 | 71.28±0.51 | 69.92±0.04 |
| | Metattack | 67.94±1.42 | 57.51±5.21 | 36.68±3.76 | 36.20±5.62 | 20.50±0.28 | **71.78±0.42** | 42.99±4.02 | 47.24±2.67 | 58.52±0.54 | **69.92±0.04** |
| | Nettack | 70.05±1.10 | 59.40±4.17 | 46.45±3.16 | 58.18±2.65 | **71.93±1.10** | 70.33±0.50 | **71.01±0.34** | 65.27±1.21 | 70.88±0.78 | 69.90±0.00 |
| Wisconsin | *clean* | 67.84±3.84 | 43.53±6.19 | 56.08±4.31 | 52.35±7.18 | 69.22±0.40 | **76.08±3.33** | 76.05±6.08 | 63.73±3.95 | 52.68±1.21 | **85.10±2.35** |
| | Random | 69.61±4.49 | 44.71±6.43 | 51.18±5.44 | 51.76±6.27 | 51.56±5.63 | 75.23±3.13 | **76.47±4.12** | 59.02±4.59 | 51.18±0.98 | **85.29±1.81** |
| | PRBCD | 67.65±5.28 | 44.71±6.72 | 55.88±4.41 | 51.37±6.67 | 52.55±5.13 | 74.60±3.14 | **75.29±4.12** | 60.39±6.61 | 50.98±0.78 | **84.90±2.33** |
| | Metattack | 64.51±5.98 | 43.53±4.09 | 50.98±4.64 | 50.59±6.06 | 52.15±5.08 | **76.67±3.92** | 74.71±4.31 | 60.39±4.79 | 51.67±1.47 | **84.90±1.76** |
| | Nettack | 70.78±6.17 | 44.71±5.32 | 55.29±5.02 | 50.00±5.70 | 53.73±5.16 | **77.65±3.92** | 75.49±3.73 | 59.02±2.97 | 51.57±1.57 | **85.10±2.00** |
| Cornell | *clean* | **63.24±9.38** | 42.97±6.78 | 51.89±6.71 | 51.08±5.19 | 59.33±1.48 | 43.78±3.51 | 54.05±7.05 | 52.97±5.82 | 45.59±0.67 | **73.78±5.68** |
| | Random | **63.24±7.27** | 37.30±4.49 | 40.00±4.95 | 49.19±4.15 | 42.97±8.10 | 43.78±5.14 | 45.68±3.51 | 54.32±6.33 | 45.49±7.72 | **73.78±5.68** |
| | PRBCD | **64.86±5.27** | 41.62±9.83 | 48.38±6.33 | 48.92±5.98 | 46.22±9.66 | 44.59±4.05 | 51.08±3.24 | 53.24±6.74 | 45.14±7.65 | **73.78±5.68** |
| | Metattack | **67.03±5.51** | 38.65±6.63 | 49.73±7.85 | 49.73±6.07 | 45.14±6.87 | 42.43±4.87 | 48.11±5.14 | 55.68±5.86 | 45.22±8.33 | **73.78±5.68** |
| | Nettack | **66.49±6.53** | 41.08±7.03 | 50.54±6.95 | 49.19±5.24 | 49.73±7.45 | 43.78±3.24 | 52.43±3.51 | 51.89±3.59 | 44.68±7.97 | **73.78±5.68** |
| Texas | *clean* | **74.32±5.16** | 57.57±5.68 | 61.35±6.63 | 57.84±5.69 | 71.08±0.34 | 72.16±3.51 | 69.73±6.26 | 68.92±5.95 | 53.60±2.67 | **77.57±4.37** |
| | Random | 72.70±4.59 | 55.41±6.97 | 55.14±5.82 | 54.59±8.18 | 56.22±5.95 | **73.51±2.16** | 64.59±2.97 | 73.51±3.24 | 53.92±3.27 | **77.03±5.30** |
| | PRBCD | **74.05±6.53** | 57.57±5.14 | 58.38±9.06 | 57.84±5.16 | 57.03±4.67 | 67.30±4.87 | 64.59±3.24 | 65.95±4.32 | 53.51±2.14 | **77.30±6.30** |
| | Metattack | **72.97±5.41** | 55.41±7.38 | 55.95±5.14 | 56.49±5.33 | 58.11±6.14 | 68.92±4.32 | 66.49±2.70 | 63.24±4.55 | 53.38±2.27 | **78.11±5.98** |
| | Nettack | **73.24±5.33** | 56.22±6.49 | 61.08±7.17 | 56.49±6.89 | 56.76±5.70 | 71.08±4.86 | 65.41±2.97 | 64.59±4.26 | 53.92±3.27 | **77.84±5.10** |
| Actor | *clean* | **35.08±1.08** | 29.25±1.21 | 24.36±1.11 | 30.72±0.72 | 30.03±0.13 | 34.37±0.69 | 33.69±0.73 | 30.02±0.91 | 26.74±0.13 | **36.56±0.93** |
| | Random | **35.15±0.78** | 27.59±1.12 | 25.64±1.02 | 30.16±1.09 | 28.36±1.95 | 25.41±0.72 | 34.19±0.59 | 28.92±1.03 | 27.09±0.68 | **36.19±0.77** |
| | PRBCD | **35.04±0.90** | 27.76±1.66 | 24.95±0.89 | 30.48±1.28 | 28.37±1.95 | 27.21±0.64 | 26.23±0.79 | 28.66±2.01 | 26.79±0.82 | **36.47±1.05** |
| | Metattack | 32.34±7.10 | 28.00±1.10 | 25.54±0.75 | 30.34±1.04 | 28.45±1.26 | 28.29±0.42 | 26.97±0.65 | 29.65±1.12 | 26.78±0.91 | **36.56±1.12** |
| | Nettack | **34.97±0.88** | 28.87±0.73 | 25.51±0.95 | 30.86±0.96 | 28.60±1.20 | 25.96±0.86 | 27.20±0.74 | 30.05±0.81 | 26.72±0.79 | **36.14±0.67** |

Table 11: Graph classification results (%); The first 4 rows are from node-level GCL methods adapted to graph-level tasks, and the next 3 rows are from graph-level models.

| Method | Proteins | DD | PTC-MR | MUTAG | Avg. rank. |
|---|---|---|---|---|---|
| MVGRL | 74.02±0.30 | 75.20±0.40 | - - | 89.20±1.30 | 5.33 |
| GraphACL | 73.50±0.70 | - - | - - | 89.40±2.00 | 5.50 |
| SimMLP | **75.30±0.10** | 78.40±0.50 | 60.30±1.10 | 87.70±0.20 | 3.63 |
| SDMG | 73.16±0.16 | 72.66±3.16 | 56.70±2.02 | **91.58±0.28** | 5.00 |
| InfoGraph | 74.44±0.40 | 72.85±1.70 | **61.70±1.40** | 89.00± 1.10 | 4.50 |
| GraphCL | 74.39±0.45 | **78.62±0.40** | - - | 86.80± 1.30 | 4.67 |
| DRGCL | 75.20±0.60 | 78.40±0.70 | - - | 89.50± 0.60 | **2.83** |
| GCN-MLP | **75.41±0.35** | 77.00±0.45 | **62.27±1.44** | **89.56± 0.85** | **2.00** |

