# OpenReview forum: "Less is More: Towards Simple Graph Contrastive Learning"
_ICLR.cc/2026/Conference — ICLR 2026 Conference Withdrawn Submission_

### Official Review · Reviewer_pexd · 2025-10-20

**Soundness:** 2
**Presentation:** 3
**Contribution:** 2
**Rating:** 2
**Confidence:** 4

**Summary:**

This paper proposes a dual-view contrastive learning framework for graph representation learning, where a GCN and an MLP respectively encode structural and feature views. The two embeddings are aligned via a simple cosine-similarity objective without requiring data augmentation or negative sampling. A learnable scalar adaptively balances the two views. Experiments on standard homophilic and heterophilic benchmarks show that the proposed model achieves competitive or superior performance with significantly lower complexity, highlighting that simplicity can rival or surpass more elaborate GCL designs.

**Strengths:**

- The paper presents a remarkably minimal yet effective framework, questioning the necessity of complex data augmentations and negative sampling in graph contrastive learning.
- It demonstrates consistent improvements over recent baselines, particularly under challenging non-homophilic settings, highlighting the method’s robustness and generality.

**Weaknesses:**

- The framework essentially reuses standard contrastive alignment (cosine similarity between two encoders), which limits its novelty.
- The analysis does not clearly disentangle the contributions of individual components such as the learnable view-weight parameter, encoder depth, or loss formulation, making it hard to attribute the observed gains.
- Although the paper emphasizes simplicity and scalability, all experiments are conducted on small benchmark graphs (e.g., Cora, Texas). Evaluations on larger real-world non-homophilous datasets [1] would be necessary to substantiate claims of efficiency and scalability.

[1] Large Scale Learning on Non-Homophilous Graphs: New Benchmarks and Strong Simple Methods, NeurIPS'21

**Questions:**

See weaknesses above.

---

> ### Author Response · Authors · 2025-11-20
> **Clarification of the novelty and contribution of our GCN-MLP framework**
>
> We sincerely thank the reviewer for the comment in Weakness 1. We would like to clarify that the novelty of our work is **not the contrastive loss itself** —we intentionally adopt a standard alignment objective. **Reusing well-established loss designs is common practice and does not diminish our contribution**.
>
> Our primary contribution is **a new principle for graph contrastive learning  (see Proposition 2 and Observation 1)**: when feature noise and structural noise are weakly correlated, their contrastive interaction (and simple linear fusion) yields stronger noise mitigation. This principle, motivated by the classical observation that the sum of two vectors cancels noise more effectively as their correlation decreases, drives our design goal: construct two views whose noise components are as uncorrelated as possible. We show that a minimal encoder pair (MLP + GCN) suffices to realize this principle and accounts for the empirical improvements reported. While the resulting GCN-MLP architecture is simple, we believe that this is a **strength** rather than a weakness, as it demonstrates **the power of our underlying principle**.

---

> ### Author Response · Authors · 2025-11-20
> **Clarification of Weakness 2**
>
> We would like to clarify that the roles of the key component of GCN-MLP **has already been examined in Sec. 5.3 in the original manuscript**. Specifically, we separately evaluate **(i) the effect of the GCN and MLP branches (Table 3) and (ii) the impact of encoder depth** by varying the number of GCN layers (Fig.3 (b)). These analyses show that neither GCN-only nor MLP-only models match the performance of GCN-MLP, underscoring the importance of combining two complementary views.
>
> Regarding the view-weight parameter $\beta$, it is not learnable in our framework; We either fix it to 0.5 for simplicity or select it via validation (step size 0.1) for potentially better performance. The results in Tables 1 and 2 of the original manuscript correspond to the **tuned values of $\beta$**. To assess sensitivity, we compare these tuned results on both heterophilic and homophilic datasets with those obtained under the default setting $\beta = 0.5$. As shown in the Table 1, the performance with fixed $\beta = 0.5$ remains reasonably close, indicating that the model is **not sensitive to the choice of $\beta$**.
>
> For the loss formulation, while **the choice of loss function is not our main contribution**; we adopt the **standard well-established cosmean loss**. We also compare  negative-sampling-free Barlow Twins loss, CCA-style losses (e.g., CCA) and negative-sampling-based InfoNCE loss on homophilic (Photo) and heterophilic (Crocodile) graphs (Table 2), showing comparable results and **confirming our model’s independence from specific loss functions**.
>
> **Table 1: Sensitiveness of GCN-MLP performance w.r.t. $\beta$**
>  Dataset    | Accuracy $(\beta)$      | $\beta = 0.5$
> |------------|-----------------------------|-------------------------|
> | Crocodile  | 66.47 ± 1.20 (0.3)          | 65.44 ± 1.20            |
> | Wisconsin  | 85.29 ± 2.19 (0.5)          | 85.29 ± 2.19            |
> | Roman      | 78.21 ± 0.39 (0.6)          | 77.98 ± 0.37            |
> | Pubmed     | 79.00 ± 0.03 (0.8)          | 78.34 ± 0.37            |
> | Photo      | 93.41 ± 0.87 (0.7)          | 92.05 ± 1.15            |
>
>  **Table 2: Sensitiveness of GCN-MLP performance w.r.t. loss functions**
>
> | Loss Function | Photo               | Crocodile            |
> |---------------|----------------------|-----------------------|
> | Barlow-twin   | 91.44 ± 1.71         | 63.72 ± 1.11          |
> | CCA           | 90.97 ± 2.20         | 63.81 ± 1.46          |
> | InfoNCE       | 93.40 ± 0.96         | 65.71 ± 0.95          |
> | Cosmean       | 93.41 ± 0.87         | 66.47 ± 1.20          |

---

> ### Author Response · Authors · 2025-11-20
> **Clarification of Weakness 3**
>
> We would like to clarify that our GCN-MLP framework **has been evaluated on large-scale** heterophilic datasets, such as **Arxiv-year (contained in your recommended reference [1]; see Table 1 and line 303 in the original manuscript for details)**. In addition, we have conducted further evaluations on **other large-scale and challenging datasets, including Roman-Empire and Amazon-Ratings**, which are commonly used as benchmarks in existing GCL methods (e.g., PolyGCL, ICLR '24; SDMG, ICML '25). These results demonstrate **the scalability and effectiveness of our method under real-world heterophilic conditions**.

---

> ### Author Response · Authors · 2025-11-25
> **Follow-up on Author Responses**
>
> Dear Reviewer pexd,
>
> We hope that the additional discussions on the novelty of our principle, the component analyses, the evaluations on larger and more challenging datasets are helpful to your assessment. If any aspect feels incomplete or could benefit from further detail, please feel free to let us know, we would gladly elaborate. If you feel that our responses resolve your concerns, we would sincerely appreciate any reconsideration of your evaluation.
>
> Thank you again for your time and for contributing to the review process.
>
> Best regards,
>
> Authors

---

> ### Comment · Reviewer_pexd · 2025-11-25
>
> I would like to thank the authors for their response and am willing to raise my score. However, given my remaining concerns about novelty, I still believe the paper is marginally below the acceptance threshold.

---

> ### Author Response · Authors · 2025-11-26
>
> We respectfully recognize the reviewer’s score of 4 concerning the novelty assessment. However, we would like to clarify that **our core intention is to let GCL return to its essence—simplicity guided by principle**. Motivated by this philosophy, we revisit the fundamental mechanism behind GCL and identify a key perspective: when feature noise and structural noise are weakly correlated, their contrastive interaction (or simple linear fusion) leads to significantly stronger noise mitigation. To the best of our knowledge, this is **the first work that explicitly formulates GCL from a noise-cancellation viewpoint.**
>
> Our GCN-MLP architecture serves as a **minimal yet effective instantiation of this principle**: the GCN captures structural information along with its associated structural noise, while the MLP isolates feature noise. By unifying these two complementary encoders through a simple design, the model naturally exploits their weak correlation. **This reflects the notion that power often emerges from simplicity**, enabling strong empirical performance despite the lightweight architecture.
>
> **We kindly hope the reviewer may reconsider this aspect after reading our clarification**, even if the final score remains unchanged.

---

### Official Review · Reviewer_fKkq · 2025-10-25

**Soundness:** 3
**Presentation:** 3
**Contribution:** 3
**Rating:** 6
**Confidence:** 4

**Summary:**

This paper presents an approach to graph contrastive learning, particularly for heterophilic graphs. Their proposed GCN-MLP model uses a standard GCN to capture the structural view and a simple MLP to capture the feature view. By contrasting these two views, which are argued to have less correlated noise sources, the model achieves strong performance without requiring any data augmentation or negative sampling, leading to significant gains in efficiency.

**Strengths:**

1. The core insight of decoupling and contrasting "feature noise" and "structural noise" is both elegant and intuitive. It provides a principled argument against the need for complex, heuristic-based data augmentations, representing a significant departure from mainstream GCL research.

2. The empirical results are compelling. The model achieves SOTA performance on several challenging heterophilic benchmarks while, as shown in Table 6, being orders of magnitude faster and more memory-efficient than its competitors.

3. The paper's clarity is one of its strongest points. The proposed GCN-MLP model is straightforward to understand and implement, which greatly enhances its potential for adoption. The authors do an excellent job of justifying the simplicity of their design choices.

4. The method shows superior robustness against adversarial attacks. This is a well-explained and significant benefit of the dual-view design, as the MLP-based view is inherently resilient to topological perturbations.

**Weaknesses:**

The main limitation of the paper, which the authors acknowledge, is its performance on certain homophilic graphs (e.g., Cora). On these graphs, where feature and structure information are highly correlated, the benefit of separating them is less pronounced, and the model can be slightly outperformed by specialized methods. While this is a reasonable trade-off, it might be worth briefly discussing whether a simple mechanism could be introduced to adaptively weigh the views based on the graph's homophily level.

**Questions:**

1.Have the authors experimented with other GNN backbones for the structural encoder? Is there a specific reason why the smoothing property of GCN and the feature-transforming nature of MLP create such an effective pairing for noise decoupling?

2.The final representation is a weighted average of the two views, controlled by a hyperparameter β. How sensitive is the model's performance to the choice of β? Figure 1 shows impressive results, but it would be helpful to understand the tuning effort required to achieve them.

3.The framework is presented for unsupervised learning. Have the authors considered its application in a semi-supervised setting? For example, could this contrastive objective serve as a powerful pre-training or regularization task to further boost performance when a few labels are available?

---

> ### Author Response · Authors · 2025-11-20
> **Response to Weakness 1**
>
> We sincerely thank you for your appreciation of our core insight behind GCN-MLP and for your **valuable suggestion in Weakness 1.**
>
> In our theoretical framework, if a node is surrounded by others belonging to the same class, then the usual aggregation is sufficient to reduce the **feature noise**, and the coupling between feature and structural noise does not bring additional benefit. However, **homophily is a local concept**; therefore, even on a homophilic dataset, there are **heterophilic nodes**, which are along class boundaries and hence difficult to classify. To evaluate whether our GCN-MLP model provides the additional benefits on such challenging cases, we focus **the evaluation on nodes with a heterophily ratio of 1, i.e., nodes whose neighbors all belong to different classes.** We report test accuracy on this subset. As shown below, our GCN-MLP consistently outperforms GraphACL, a strong contrastive baseline widely recognized for its performance on homophilic datasets.
>
> **Table: Test accuracy on heterophilic nodes.**
>
> | Model     | Cora            | Citeseer        | Pubmed         |
> |-----------|------------------|------------------|-----------------|
> | GraphACL  | 36.31 ± 0.01     | 31.06 ± 0.87     | 43.43 ± 0.03    |
> | **GCN-MLP** | **39.11 ± 0.65**     | **32.95 ± 0.13**     | **54.07 ± 0.57**    |
>
> The observation not only aligns with our theoretical analysis but also **offers insight into how our model may contribute to homophilic datasets**. For future work, it motivates exploring whether the feature fusion parameter $\beta$ can be made node-adaptive. Inspired by [1], a promising extension is to allow $\beta$ (which controls view fusion) to vary across nodes based on the estimated noise correlation between structural and feature views. This could further enhance the model's ability to balance structural and feature noise cancellation in a data-dependent manner.
>
> [1] Zhang et al, ApeGNN: Node-wise adaptive aggregation in GNNs for recommendation, WWW 2023.

---

> ### Author Response · Authors · 2025-11-20
> **Response to Question 1, 2 and 3**
>
> We appreciate the reviewer’s valuable Questions 1–3 and address them point by point.
>
> **Q1. Reason behind the effectiveness of noise decoupling and other GNN backbones testing.**
>
> Conceptually, the contribution of GNN aggregation is that its **smooths** or, more precisely, **averages out** feature noise, as established in **Proposition 2** and the accompanying discussion. As a result, a GCN effectively replaces the original feature noise with structural noise. Consequently, the features produced by the GCN and those produced by the MLP carry noise of different natures. Unless the original features and the graph construction are strongly correlated (as in homophilic settings), the feature noise and structural noise become decoupled and only weakly correlated.
>
> Therefore, **any GNN model that exhibits a similar averaging effect, such as GAT, might be used in place of GCN**. The result in the **Table 1** supports this perspective.
>
> **Table 1: Different GNN backbones as structural encoder.**
>
> | Dataset   | GCN-MLP          | GAT-MLP         |
> |-----------|-------------------|------------------|
> | Crocodile | 66.47 ± 1.20      | 67.12 ± 0.89     |
> | Wisconsin | 85.29 ± 2.19      | 84.12 ± 2.05     |
> | Roman     | 78.21 ± 0.39      | 77.07 ± 0.45     |
> | Pubmed    | 79.00 ± 0.03      | 79.52 ± 0.19     |
> | Photo     | 93.41 ± 0.87      | 92.12 ± 0.83     |
>
> **Q2. Sensitiveness of GCN-MLP' performance w.r.t. the choice of $\beta$.**
>
> There are two options for choosing $\beta$: it can be fixed at 0.5 for convenience, or selected via validation with a step size of 0.1, which may yield better performance. The results reported in Tables 1 and 2 of the original manuscript correspond to the tuned values of $\beta$. To assess sensitivity, we compare these tuned results on both heterophilic and homophilic datasets with those obtained under the default setting $\beta = 0.5$. As shown in the **Table 2** below, the performance with fixed $\beta = 0.5$ remains reasonably close, indicating that **the GCN-MLP is not sensitive to the choice of $\beta$**.
>
>  **Table 2: Sensitiveness of GCN-MLP performance w.r.t. $\beta$**
>
> | Dataset   | Accuracy $(\beta)$ | $\beta=0.5$  |
> |-|-|-|
> | Crocodile | 66.47 ± 1.20 (0.3)   | 65.44 ± 1.20  |
> | Wisconsin | 85.29 ± 2.19 (0.5)   | 85.29 ± 2.19  |
> | Roman | 78.21 ± 0.39 (0.6)   | 77.98 ± 0.37 |
> | Pubmed  | 79.00 ± 0.03 (0.8)   | 78.34 ± 0.37 |
> | Photo | 93.41 ± 0.87 (0.7)   | 92.05 ± 1.15 |
>
>
> **Q3. Semi-supervised setting of GCN-MLP**
>
> We would like to clarify that our GCN-MLP framework is designed as an unsupervised contrastive framework, where the objective focuses **purely on learning robust and discriminative feature representations without relying on any label information**. Therefore, it is a pre-trained model, and the output representation can be used for downstream tasks, which is node classification in our case.
>
> For node classification, we just follow **common practice** in representation learning and **consist with the GCL evaluation protocol (e.g., BGRL, GraphACL and PolyGCL)**. Specifically, we adopt a linear evaluation setup in which a simple linear classifier is trained on top of the frozen embeddings using full label information. This protocol isolates the encoder’s representational ability from downstream supervised effects and provides a fair and widely used measure of the quality of the learned representations.
>
> Regarding the semi-supervised setting, it suffices to modify the downstream task by providing partial labels. **Our framework naturally supports scenarios with limited label availability**. When only a small portion of labels is provided, these labels can be directly incorporated into the linear classifier training phase to obtain a semi-supervised variant. As expected, using fewer labels leads to a gradual performance drop; however, our preliminary experiments (shown in **Table 3**) show that the learned representations still outperform strong baselines (e.g., GraphACL) on the Texas, even when the training label ratio is progressively reduced (i.e., 1, 0.8, 0.5, and 0.1). We attribute this robustness to the core design of GCN-MLP: by contrasting and canceling structural noise and feature noise, the framework produces **noise-resilient and discriminative representations that remain effective under scarce labels**.
>
>  **Table 3: Performance on semi-supervised setting.**
>
> | Training ratio | 1             | 0.8           | 0.5           | 0.1             |
> |----------------|----------------|----------------|----------------|------------------|
> | GraphACL       | 71.62 ± 7.12   | 67.03 ± 5.06   | 61.84 ± 6.67   | 54.59 ± 10.49    |
> | **GCN-MLP**   | **78.38 ± 4.68**   | **77.30 ± 5.16**   | **71.62 ± 5.44**   | **60.81 ± 10.42** |

---

> ### Author Response · Authors · 2025-11-25
> **Follow-up on Author Responses**
>
> Dear Reviewer fKkq,
>
> Thank you again for your thoughtful and encouraging review of our paper. Your recognition of the core idea behind the dual-view noise decoupling and your constructive questions were extremely helpful for improving our work.
>
> We submitted our responses on 20 Nov and would like to gently check whether any further clarification from our side might be useful for your evaluation. If you feel that these responses adequately address the concerns you listed in your review, we would sincerely appreciate any reconsideration of your assessment at your discretion.
>
> Please feel free to let us know if additional details or experiments would be helpful. We would be more than happy to provide them promptly.
>
> Thank you very much for your time and your feedback to the review process.🌷
>
> Warm regards,
>
> Authors

---

> ### Comment · Reviewer_fKkq · 2025-11-25
>
> Having reviewed the authors’ response, I confirm that my concerns have been resolved and no additional inquiries arise. I wish to keep my score.

---

### Official Review · Reviewer_UouA · 2025-10-27

**Soundness:** 3
**Presentation:** 3
**Contribution:** 1
**Rating:** 2
**Confidence:** 5

**Summary:**

This work proposes a GCL method that aligns GCN with node-attribute based MLP and integrates the features of the two channels (i.e., GCN encoder and MLP encoder)  as the final node representation. The performance of the proposed method  is evaluated on node classification task for homophilious/heterophilic graphs, and the robustness of the model against intentional  and random structural perturbations is assessed.

**Strengths:**

The paper is clearly written, easy to follow and  technically correct. The experiments can be reproducible.

**Weaknesses:**

*  high similar works SimMLP (arxiv:2412.03864) and DGCL (Briefs in bioinformatics, 2024) have been published a year ago, which also align GNN-structured view with node-attribute invariant MLPs.
* Some claims such as "uncover a simple yet effective principle for GCL:..." have been the common ground in GNN communities for years, e.g., earlier study of heterophilic graph learning by Zhu et al. (AAAI'20) reveals the advantages of MLPs on heterophilic graphs.
* Common tasks including link prediction/ graph classification / node clustering are missing, which can be more sensitive to structural information.

**Questions:**

* The robustness of the proposed GCL model is conducted on attacking structure. What if node attributes  are adversarially attacked, e.g. using metattack?
* Besides the previous works mentioned in "Weakness", GECL(Xiao et al, 2024) also adopts the similar framework: Aligning MLPs encoder with GNNs. How your model distinguish from the previous ones needs to be clarified.
* What is the role of weighted sum of two channels? Do you use the similar feature fusion approach for the baselines?

---

> ### Author Response · Authors · 2025-11-20
> **Clarification of the Differences Between GCN-MLP and SimMLP, DGCL, and GraphECL**
>
> We sincerely thank the reviewer for the helpful comment in **Weakness 1 and Question 2**. We would like to clarify that GCN-MLP is **fundamentally different from SimMLP, DGCL, and GraphECL in terms of principles, motivations, and architectures**.
>
> First, DGCL (Briefs in bioinformatics, 2024) employs dual GNNs (i.e., GAT and GIN) to extract heterogeneous structural features. This design philosophy is entirely different from GCN-MLP encoder, which pairs a GCN with and an MLP. The GCN captures structural features together with their inherent structural noise, while the MLP isolates feature-level noise. This yields two complementary views for contrastive learning.
>
> Second, GCN-MLP is also distinct from SimMLP (arxiv:2412.03864) and GraphECL in both motivation and contrastive formulation. SimMLP and GraphECL distill knowledge from a teacher GNN into a student MLP trained solely on node features, with the aim of integrating rich structural information into the MLP. Only the MLP outputs are used for downstream tasks, with the primary goal of accelerating inference by replacing GNN computation.
>
> In contrast, GCN-MLP framework is **not a distillation model**. Instead, we present **a new underlying principle for graph contrastive learning**: feature noise and structural noise are weakly correlated, enabling stronger mitigation when contrasted and combined. This is motivated by the classical observation that noise mitigation in the sum of two vectors is stronger when the vectors are less correlated. Therefore, our objective is to construct two feature views whose associated noise components are as uncorrelated as possible.
>
>
> To achieve this, GCN-MLP serves as a simple yet effective instantiation of our principle. The GCN branch captures structural features together with their inherent structural noise, while the MLP isolates feature-level noise. This yields two complementary views whose weakly correlated noise components naturally facilitate noise cancellation through contrastive learning and linear combination, resulting in cleaner and more discriminative features for node classification, particularly on challenging heterophilic graphs.
>
> In addition, GCN-MLP achieves remarkable performance on heterophilic graphs (e.g., Wisconsin and Roman) compared with SimMLP and GraphECL, as shown in **Table 1**, while requiring less training time and comparable memory usage. These results further confirm the effectiveness and effiency of our proposed GCN-MLP framework.
>
> In summary, although other works have utilized GNN and MLP in GCL, their approaches and motivations are **fundamentally different** from ours. Our work is the **first** to identify and leverage the weak correlation between feature noise and structural noise in a GCN-MLP framework for effective graph contrastive learning. While the resulting GCN-MLP architecture is simple, we believe that this is a **strength** rather than a weakness, as it demonstrates the power of our underlying principle.
>
> Based on your suggestion, we have **clarified these differences in the revised manuscript by updating the Related Work section**. Specifically, we now explicitly cite SimMLP and DGCL, and provide a discussion highlighting how our approach differs from these methods in terms of architectural design, motivation, and underlying principles.
>
>
> **Table 1: Accuracy / Training time (s) / Memory usage (MiB) comparison.**
>
> | Model     | Wisconsin                 | Roman                     |
> |-----------|----------------------------|----------------------------|
> | SimMLP    | 56.57 ± 2.59 / 0.008 / 79  | 64.72 ± 0.07 / 0.035 / 1159 |
> | GraphECL  | 79.41 ± 2.19 / 0.27 / 336  | 45.05 ± 1.57 / 60.58 / 2306 |
> | GCN-MLP   | **85.29 ± 2.19 / 0.007 / 588** | **78.21 ± 0.39 / 0.097 / 5850**|

---

> ### Author Response · Authors · 2025-11-20
> **Clarification the novelty and contribution of our GCN-MLP framework**
>
> We sincerely thank the reviewer for the helpful comment in **Weakness 2**. We would like to clarify that **the novelty and contribution of our GCN-MLP framework is in uncovering an underlying principle for graph contrastive learning from a new perspective**: feature noise and structural noise are weakly correlated **(See Observation 1 and Theorem 1 in Appendix B)**, enabling stronger mitigation when contrasted and combined. This insight is motivated by an observation in vector analysis: the noise mitigation in the sum of two vectors becomes stronger when the vectors are less correlated. Therefore, our goal is to construct two feature views whose associated noise components are as uncorrelated as possible.
>
> **The primary contribution of our paper is to discover this principle for graph GCL (see Proposition 2 and Observation 1)**, which naturally leads to a simple yet highly effective GCN-MLP architecture. Specifically, the GCN captures structural information together with the structural noise induced by message passing on graphs, while the MLP isolates node-feature noise by operating purely in the feature space. These two encoders form a pair of complementary views whose weakly correlated noise components naturally facilitate noise cancellation during contrastive learning and through their linear combination, resulting in cleaner, more discriminative representations for node classification.
>
> Importantly, the success of GCN-MLP **does not stem from either the GCN or the MLP alone, but rather from their interaction**, which arises from GCN-MLP encoder pairs that generate views with distinct but weakly correlated noise, **numerically supported by Ablation study in Sec. 5.3**. It is precisely this interaction enables GCN-MLP to outperform more complex contrastive frameworks while retaining its simplicity.
>
> Regrading Zhu et al (AAAI'20), their finding is **not directly related to our work**. It showed that an MLP can outperform GCNs on heterophilic graphs in supervised settings, as MLP avoids propagating misleading structural information. Our GCN-MLP **differs in two key ways**: (i) our framework is in unsupervised manner, (ii) the effectiveness of GCN-MLP does not come from MLP alone, but from the construction of two views whose noise components are weakly correlated, enabling noise cancellation. This principle is **fundamentally different from the supervised phenomenon observed by Zhu et al**.

---

> ### Author Response · Authors · 2025-11-20
>
> **Question 1. Adversarially attack for the node attributes**
>
> We would like to clarify that the robustness of GCN-MLP **has been** evaluated under adversarial attacks targeting both the graph structure and node features. Specifically, we respectfully refer the reviewer to **Table 5 in the original manuscript**, where we report results under a white-box attack that jointly perturbs the graph topology and node attributes **(see line 410 in the original manuscript)**. For example, on the photo dataset, the accuracy of SDMG '25 drops to 3.11% (from 94.10%) under the PGD attack, while our GCN-MLP maintains a 90.56 \% accuracy.
>
> **Question 3. The role of weighted sum of two channels**
>
> The weighted sum of the two channels serves as a **simple feature-fusion mechanism** that aggregates the complementary information captured by the GCN and MLP branches. Since the two channels focus on a pair of complementary views, the weighted sum provides a simple and effective way to fuse their information. **This fusion strategy is standard in GCL methods**, for example, PolyGCL fuses low-pass and high-pass polynomial-filter views using a weighted combination, and GraphACL combines two encoder views with finetuned coefficients. **Therefore, our use of a weighted sum is fully aligned with existing GCL practices and does not introduce any unfair advantage.**
>
> **Weakness 3. Broader application beyond node classification**
>
> While most GCL methods target node-level representation learning and do not provide a straightforward graph-level extension, we assess GCN-MLP's generality by applying a simple, non‑parametric readout (MeanPooling) to obtain graph-level embeddings. We evaluate this configuration on two standard graph-classification benchmarks, Proteins and DD, and compare against recent graph-level GCL models such as GraphCL and DRGCL as well as strong node-level baselines adapted to the graph-level setting (e.g., GraphACL). **GCN-MLP achieves competitive performance compared with node-level baselines and yields results on par with specialized graph-level contrastive methods**, demonstrating that our proposed GCN-MLP framework generalizes effectively beyond node-level tasks.
>
> **Table: Graph classification results (%). The first 4 rows are from node-level GCL methods adapted to graph-level tasks, and the next 3 rows are from graph-level models.**
>
> | Method     | Proteins        | DD              |
> |------------|------------------|------------------|
> | MVGRL      | 74.02 ± 0.30     | 75.20 ± 0.40     |
> | GraphACL   | 73.50 ± 0.70     | --               |
> | SimMLP     | 75.30 ± 0.10     | 78.40 ± 0.50     |
> | SDMG       | 73.16 ± 0.16     | 72.66 ± 3.16     |
> | InfoGraph  | 74.44 ± 0.40     | 72.85 ± 1.70     |
> | GraphCL    | 74.39 ± 0.45     | 78.62 ± 0.40     |
> | DRGCL      | 75.20 ± 0.60     | 78.40 ± 0.70     |
> | **GCN-MLP** | **75.41 ± 0.35**     | **77.00 ± 0.45**     |

---

> ### Author Response · Authors · 2025-11-25
> **Follow-up on Author Responses**
>
> Dear Reviewer UouA,
>
> In our responses above, we tried to carefully address each of the issues you highlighted. We would like to check whether our clarifications resolve the concerns you raised. If there is anything that remains unclear or if any part of our explanation could benefit from further detail, we are fully committed to providing any additional details that may help, and sincerely hope our clarifications are helpful to any reconsideration of your evaluation. :)
>
> Thank you again for your time and for contributing to the review process.
>
> Kind regards,
>
> Authors

---

> > ### Comment · Reviewer_UouA · 2025-11-27
> >
> > Thank you to the authors for their detailed responses. Some of my concerns have been resolved, and as a result I have raised my score. However, two major issues remain that still prevent me from supporting acceptance in the current form:
> >  1) The experimental validation of adversarial robustness and graph-level classification performance is still limited to only two datasets. Given the well-known dataset-specific quirks in graph benchmarks, claims of superior robustness and effectiveness can only be considered sufficiently substantiated with results on at least four diverse datasets. I strongly encourage the authors to include additional standard benchmarks in the final version.
> >  2)Although the authors correctly point out differing training objectives, the core feature-learning paradigm (constructing and aligning multiple augmented views to extract uncorrrelated information) is conceptually highly similar to SimMLP and related contrastive works. The consistent underperformance against SimMLP on DD may contradict the central claim that integrating uncorrrelated noisy views leads to better noise mitigation and more robust representations. Without either new experiments showing superiority on additional graphs, or a clearer theoretical/empirical explanation of this discrepancy, the claimed advantage remains unconvincing.

---

> ### Author Response · Authors · 2025-11-29
> **Response to additional two questions**
>
> We sincerely thank the reviewer for explicitly highlighting **two questions** in the current version, which we address point by point below.
>
> **Q1. Validation of adversarial robustness and graph-level classification performance limited to two datasets; require additional benchmarks. (at least 4 diverse datasets)**
>
> Following your suggestions, we have added additional standard benchmarks (covering up to four diverse datasets) for both adversarial robustness and graph-level classification. **These new results are reported in Table 7 (line 503) and Table 11 (line 1173 in the Appendix) of the revised manuscript, respectively**. For convenience, we also relist the corresponding tables below.
>
> Regarding **Table 1 (white-box adversarial attacks)**, the updated results clearly demonstrate the **strong robustness of our method**, particularly under the more challenging setting where both node attributes and the graph topology are jointly perturbed. This robustness is especially evident on heterophilic graphs, where structural perturbations are generally more harmful.
>
> For **Table 2 (graph-level classification)**, we **have added two additional standard benchmarks (PTC-MR and MUTAG) and also reported the average ranking** computed as the arithmetic mean of the valid per-dataset ranks (excluding missing entries). Although our method is not the top performer on every dataset, it achieves the lowest overall average rank, indicating the **best overall performance across all benchmarks**. Though not specifically designed for graph classification, GCN-MLP delivers competitive results compared with node-level baselines and performs on par with specialized graph-level contrastive methods, demonstrating that the proposed framework generalizes effectively beyond node-level tasks.
>
>  **Table 1: White-box attack robust accuracy (\%) under 15\% perturbation for node classification.**
>
> | Dataset | Attack | FROND| Res-GRACE | GraphACL | PolyGCL | LOHA   | EPAGCL   | SDMG    | GCN-MLP   |
> |-|-|-|-|-|-|-|-|-|-|
> | Photo| clean| 92.03±1.27 | 92.50±0.17 | 93.31±0.19 | 91.45±0.35 | 86.46±0.41 | 93.05±0.23 | 94.10±0.20 | **93.41±0.88** |
> |          | PGD |14.18±5.16 | 45.46±2.05 | 30.94±3.62 | 22.93±2.73 | 59.35±0.43 | 7.71±1.41 | 3.11±1.11 | **90.56±0.54** |
> |   | PRBCD |15.08±7.52|40.72±3.70 | 28.56±1.16 |15.86±1.99  | 52.82±3.45 | 3.69±1.35  | 21.35±11.31 | **75.42±0.69**|
> | Texas | clean| 74.86±3.21| 54.59±5.51 | 71.08±0.34 | 72.43±4.86 | 69.73±6.26| 68.92±4.05 | 53.60±2.67 |**78.38±4.68**|
> | | PGD| 69.46±7.16 | 28.85±8.55| 21.62±7.20 | 56.76±16.22 |32.70±20.17 | 48.65±17.72 | 27.57±22.67 | **75.68±6.28** |
> | | PRBCD | 65.14±3.91| 26.49±11.22| 23.78±8.99| 53.51±13.23| 37.57±15.17| 46.22±16.24| 15.68±16.12| **67.03±5.10** |
> | Wisconsin | clean| 67.84±3.84| 52.35±7.18| 69.22±0.40|76.08±3.33| 76.05±6.08| 63.73±3.95| 52.68±1.21|**85.10±2.35**|
>  | | PGD| 62.16±6.01| 30.45±6.89| 8.04±3.75 | 66.27±6.06| 66.86±7.52| 18.43±10.57| 15.29±17.60| **80.98±4.64**|
>  || PRBCD | 57.06±5.71| 27.56±8.36 | 11.37±3.56| 60.59±5.65| 57.06±7.36 | 33.92±11.20 | 18.24±12.15 | **73.53±4.04**|
> | Cornell | clean| 63.24±9.38| 51.08±5.19| 59.33±1.48| 43.78±3.51| 54.05±7.05 | 52.97±5.82| 45.59±0.67 | **73.78±5.68**|
>  || PGD| 44.36±8.34| 30.58±6.13| 29.46±10.23 | 38.65±6.17 | 48.11±5.10 | 20.00±9.98 | 39.46±11.67 | **53.78±8.92** |
> || PRBCD| 43.81±8.00| 28.13±5.47| 37.03±8.74| 36.76±7.47| 44.05±7.65 | 20.54±13.20 | 13.24±8.24| **52.16±9.52** |
>
> **Table 2: Graph classification results (%). The first 4 rows are from node-level GCL methods adapted to graph-level tasks, and the next 3 rows are from graph-level models.**
>
> | Method | Proteins | DD| PTC-MR | MUTAG | Avg. rank |
> |-|-|-|-|-|-|
> | MVGRL | 74.02±0.30 | 75.20±0.40 | - - | 89.20±1.30     | 5.33      |
> | GraphACL| 73.50±0.70 | - -| - - | 89.40±2.00     | 5.50      |
> |   SimMLP    | 75.30±0.10       | 78.40±0.50      | 60.30±1.10     | 87.70±0.20     | 3.63      |
> |   SDMG      | 73.16±0.16       | 72.66±3.16      | 56.70±2.02     | 91.58±0.28     | 5.00      |
> |   InfoGraph | 74.44±0.40       | 72.85±1.70      | 61.70±1.40     | 89.00±1.10     | 4.50      |
> |  GraphCL   | 74.39±0.45       | 78.62±0.40      | - -            | 86.80±1.30     | 4.67      |
> |   DRGCL     | 75.20±0.60       | 78.40±0.70      | - -            | 89.50±0.60     | 2.83      |
> | **GCN-MLP** | **75.41±0.35**   | **77.00±0.45** | **62.27±1.44**  | **89.56±0.85**  | **2.00** |

---

> ### Author Response · Authors · 2025-11-29
> **Response to additional two questions**
>
> **Q2. Difference between ours and SimMLP and related contrastive works & new experiments on addtional graphs**
>
> We would like to clarify that our contrastive formulation is **fundamentally different from SimMLP and mainstream GCL methods such as PolyGCL and GraphACL**. As stated in the original SimMLP paper (See line 14 in the abstract), SimMLP’s core objective is to inject structural information into MLPs by aligning a graph-context-aware GNN with a neighborhood-independent MLP. **Its purpose is therefore architectural enhancement of MLPs, rather than noise modeling**.
>
>
> In contrast, we formulate GCL from a **noise-cancellation perspective**: feature noise and structural noise are weakly correlated, enabling stronger mitigation when contrasted and combined. **To the best of our knowledge, this novel perspective is not explored in GCL literature**. We **theoretically establish this principle in Observation 1 and Theorem 1 (Appendix B)**. Their weak correlation ensures that contrastive interaction (and simple linear fusion) achieves stronger noise mitigation. This insight follows a basic vector-analysis principle: the less correlated two vectors are, the more effectively their noise cancels when combined. **To our knowledge, ours is the first work to explicitly frame GCL from this noise-cancellation viewpoint, rather than from a multi-view augmentation paradigm. As Reviewer fKkq noted, this represents a significant departure from mainstream GCL research.**
>
> Mainstream GCL methods (e.g., PolyGCL and GraphACL) rely on constructing multiple graph views (typically local views and global views) using polynomial filters or one-/two-hop neighborhood contexts. **Our goal is different: we intentionally avoid complex view construction and instead return to a simple, principled mechanism driven by noise-cancellation**. The resulting GCN-MLP architecture provides a minimal yet effective instantiation of this principle: the GCN captures structural signal plus structural noise, while the MLP isolates feature-level variations. Their weak correlation makes the contrastive interaction particularly effective, yielding strong empirical performance despite the lightweight design.We believe this reflects the modern notion that **simplicity can be exceptionally powerful**.
>
> Regarding the reviewer’s concern about DD, we acknowledge that GCN-MLP does not outperform SimMLP on this specific dataset. **However, as shown in Table 2, our method achieves superior results on three other standard graph-classification benchmarks (Proteins, PTC-MR, MUTAG)**. For node classification, GCN-MLP shows notably strong performance on heterophilic graphs (e.g., Wisconsin and Roman), where SimMLP typically struggles (Table 3), while also requiring less training time and comparable memory usage. **These additional evaluations provide a broader picture and confirm the effectiveness and efficiency of our proposed framework beyond any single dataset.**
>
>
>
> **Table 3: Accuracy / Training time (s) / Memory usage (MiB) comparison.**
>
> | Model     | Wisconsin                 | Roman                     |
> |-----------|----------------------------|----------------------------|
> | SimMLP    | 56.57 ± 2.59 / 0.008 / 79  | 64.72 ± 0.07 / 0.035 / 1159 |
> | **GCN-MLP**   | **85.29 ± 2.19 / 0.007 / 588** | **78.21 ± 0.39 / 0.097 / 5850** |

---

### Official Review · Reviewer_EBbx · 2025-11-01

**Soundness:** 3
**Presentation:** 3
**Contribution:** 4
**Rating:** 6
**Confidence:** 2

**Summary:**

The paper focuses on heterophilic graph contrastive learning. Motivated by the increasing complexity of existing methods, the authors propose a simple yet effective model. Leveraging the natural low correlation between node features and graph structure in heterophilic graphs, the model uses GCN and MLP to create two complementary views for contrastive learning. The GCN-MLP model achieves competitive representation performance without any augmentation and at low cost.

**Strengths:**

i. This model adopts a straightforward design, leveraging the characteristics of heterophilic graphs. From a practical standpoint, the motivation is clear and reasonable. The effort to solve the problem with a simpler model is appealing.
ii. The proposed GCN-MLP model reduces computational costs while maintaining competitive representation performance. The experimental results clearly support this advantage.

**Weaknesses:**

i. The relationship between the two main components, GCN and MLP, should be described more clearly. It is important to emphasize that the proposed method is not simply a combination of existing architectures.
ii. Some statements lack rigor:
a) On p.2, the phrase “akin to a law-of-large-numbers effect” is not rigorous and should be rephrased in a more formal manner.
b) All symbols should be clearly defined before being used. On p.3, the notation R^F introduces F without explanation.
iii. Parameters described as empirical require further justification. For example, L. If the proposed model is not only a simple combination of MLP and GCN, the reuse of this empirical setting (L=1) should be justified. Additionally, experiments should be conducted to demonstrate the impact of this parameter on the performance.
iv. Minor presentation issue: Sentences such as “This completes our brief review of GCL.” should be removed, as they do not provide any substantive information.
v. In the theoretical analysis (Observation 1), the term “average correlation” is repeatedly used. However, it lacks a formal definition or explanation in the main text. Although it is briefly mentioned in the appendix (“if we use E^k as a measure of the average correlation...”), a formal definition would help clarify its meaning .

**Questions:**

i. In the experiments, the proposed model shows competitive representation performance without relying on data augmentation or negative sampling. Could the authors provide experimental analysis to verify whether using these techniques would further improve performance, or if the model already reaches its optimal capacity without these techniques?
ii. Could the authors consider including an additional experiment to evaluate the effectiveness of the GCN-MLP model on downstream tasks beyond node classification, to further demonstrate its performance and broader applicability?

---

> ### Author Response · Authors · 2025-11-20
> **Clarification of Weaknesses i–v**
>
> We sincerely thank you for appreciating the core insight behind GCN-MLP and for your **valuable suggestion** in Weakness i on further **emphasizing the novelty and motivation**, your comments on vague statements and presentation issues in Weakness ii, iii, iv and v.
>
> **Weakness i**
>
> We would like to clarify that **the contribution of GCN-MLP is not simply a combination of existing architectures (i.e., GCN and MLP) but uncovering a new underlying principle of graph contrastive learning**: when feature noise and structural noise are weakly correlated, their contrastive interaction (and simple linear fusion) yields stronger noise mitigation. This insight, motivated by the classical observation that the sum of two vectors cancels noise more effectively as their correlation decreases, drives our design goal: construct two views whose noise components are as uncorrelated as possible. The GCN-MLP architecture is a simple yet effective instantiation of this principle, where the GCN encodes structural information with its associated structural noise, and the MLP isolates feature noise. We emphasize that the novelty lies in the principled understanding of noise interactions, which explains why such a minimal encoder pair is sufficient to achieve the empirical improvements we report. **Based on your suggestion, we have emphasized it in the introduction in the revised manuscript**.
>
> **Weakness ii and iv**
>
> Thanks for your suggestion. For (a), we have revised the phrase *akin to a law-of-large-numbers effect* as **aggregating features across nodes of the same class reduces the variance of noisy observations, since averaging multiple samples provides a more stable estimate of the underlying class prototype**. In terms of (b), $F$ represents the feature dimension. We have made them clear in the revised manuscript.
>
> In addition, we **have removed** the sentences (i.e., this completes our brief review of GCL) in the revised manuscript.
>
> **Weakness iii. Parameter $L$ justification**
>
> We choose a small $L=1$ or $L=2$ for two reasons. (1) This aligns with common practice in the GNN literature: using a large $L$ often leads to over-smoothing, which can make the resulting representations less informative for classification. (2) We have empirically verified that a small $L$ is already sufficient for our purpose of decoupling feature and structural noise (see Fig. 2 and the accompanying discussion for details).
>
> For example, on the Cornell dataset, the accuracy drops to 50.54\% when $L = 8$ and remains at a similar level for $L = 12$ and $L = 16$ (50.81\%), indicating that over-smoothing kicks in for large $L$. A similar pattern is observed on the Roman dataset, where the accuracy decreases to approximately 50.5\% for $L = 8, 12, 16$.
>
> **Weakness v. Clear clarification of $E^k$ in Observation 1**
>
> Thanks for your suggestion. We have made the definition of $E_{k}$ clear **in the revised manuscript (See line 237)** and explained its **high-level intuition of noise cancellation within Observation 1** in order to avoid unnecessary notation. The corresponding **rigorous formulation and full proof are presented in Theorem 1 in Appendix B**, ensuring that readers can first grasp the idea and then refer to the appendix for the formal details.

---

> ### Author Response · Authors · 2025-11-20
> **Response to Question 1 and 2**
>
> We sincerely thank the reviewer for the valuable question 1 and 2, and we address them point by point.
>
> **Q1. Additional techniques (i.e., data augmentation or negative sampling) testing**
>
> We further evaluate GCN-MLP by incorporating additional operations—**data augmentation (e.g., edge removal and node-feature masking) and negative sampling** using the InfoNCE objective as in GRACE, on both homophilic and heterophilic datasets. **Table 1** shows that these techniques yield performance comparable to our original model, indicating that **GCN-MLP already operates close to its optimal capacity without relying on either augmentation or negative sampling**.
>
> This aligns with **the principle of Occam’s Razor, which advocates for simpler architectures when possible to avoid unnecessary model complexity**. Our augmentation-free and negative-sample-free design remains lightweight, effective, and efficient while maintaining strong performance. Following your suggestion, we **have included these results in Sec. 5.3 of the revised manuscript**.
>
> **Table 1: Comparison of GCN-MLP with its additional techniques.**
>
> | Model                 | Crocodile       | Wisconsin       | Roman           | Pubmed         | Photo          |
> |-----------------------|------------------|------------------|------------------|------------------|------------------|
> | GCN-MLP              | 66.47 ± 1.20     | 85.29 ± 2.19     | 78.21 ± 0.39     | 79.00 ± 0.03     | 93.41 ± 0.87     |
> | GCN-MLP(+ Aug.)       | 65.56 ± 1.01     | 85.49 ± 2.80     | 78.25 ± 0.49     | 79.05 ± 0.22     | 93.48 ± 0.82     |
> | GCN-MLP(+ Neg.)       | 65.71 ± 0.95     | 84.92 ± 3.26     | 78.16 ± 0.45     | 78.67 ± 0.13     | 93.40 ± 0.96     |
> | GCN-MLP(+ Aug. & Neg.) | 65.71 ± 0.55    | 85.69 ± 4.96     | 78.20 ± 0.49     | 78.58 ± 0.04     | 93.46 ± 0.97     |
>
> **Q2. More downstream task beyond node classification**
>
> While most GCL methods target node-level representation learning and do not provide a straightforward graph-level extension, we assess GCN-MLP's generality by applying a simple, non‑parametric readout (MeanPooling) to obtain graph-level embeddings. We evaluate this configuration on two standard graph-classification benchmarks, Proteins and DD, and compare against recent graph-level GCL models such as GraphCL and DRGCL as well as strong node-level baselines adapted to the graph-level setting (e.g., GraphACL). **GCN-MLP achieves competitive performance compared with node-level baselines and yields results on par with specialized graph-level contrastive methods**, demonstrating that our proposed GCN-MLP framework generalizes effectively beyond node-level tasks. Based on your suggestion, we **have provided this graph classification results (**Table 2**) in the revised manuscript.**
>
> **Table 2: Graph classification results (%). The first 4 rows are from node-level GCL methods adapted to graph-level tasks, and the next 3 rows are from graph-level models.**
>
> | Method     | Proteins        | DD              |
> |------------|------------------|------------------|
> | MVGRL      | 74.02 ± 0.30     | 75.20 ± 0.40     |
> | GraphACL   | 73.50 ± 0.70     | --               |
> | SimMLP     | 75.30 ± 0.10     | 78.40 ± 0.50     |
> | SDMG       | 73.16 ± 0.16     | 72.66 ± 3.16     |
> | InfoGraph  | 74.44 ± 0.40     | 72.85 ± 1.70     |
> | GraphCL    | 74.39 ± 0.45     | 78.62 ± 0.40     |
> | DRGCL      | 75.20 ± 0.60     | 78.40 ± 0.70     |
> | **GCN-MLP** | **75.41 ± 0.35**     | **77.00 ± 0.45**     |

---

> ### Author Response · Authors · 2025-11-25
> **Follow-up on Author Responses**
>
> Dear Reviewer EBbx,
>
> Thank you again for your detailed and valuable review of our submission. We submitted our responses on 20 Nov and believe we have addressed your insightful suggestions. We sincerely hope that these revisions fully satisfy the concerns you raised. If you find that our responses adequately resolve the issues, we would be very grateful if you could reconsider your assessment at your discretion.
>
> If any further details or data would help your evaluation, we would be glad to provide them immediately.
>
> Thank you for your continued effort and valuable feedback.🌻
>
> Best regards,
>
> Authors

---

### Author Response · Authors · 2025-11-29
**Summary for AC**

Dear AC,

We have addressed **all the concerns and questions raised by both Reviewer fKkq and EBbx**, who acknowledge the core insight of our framework and recognize its novelty and significance to the GCL field. Our work formulates GCL from a **fundamentally new perspective**, which **Reviewer fKkq highlights as a meaningful departure from mainstream GCL approaches**. Both express support for the acceptance of our submission.

We have also addressed most of the concerns raised by Reviewer UouA, particularly those regarding similarity to related works (e.g., SimMLP), by clearly clarifying the differences in principles, motivations, architectures, and empirical performance. Reviewer UouA acknowledged our clarification and stated on **11:37, 27 Nov** that he has **raised his score**. In addition, he is very explicit regarding his remaining concerns:

 **(1)** *We should have **four** datasets for adversarial robustness and graph-level classification.* Following this advice, we **have incorporated four diverse benchmark datasets for both tasks** in the revised manuscript.

**(2)** *We should to explain our novelty in contrast to both SimMLP and mainstream GCL methods*. We **explicitly explain that the fundamental contribution** of our work is to propose a new principle of GCL as noise decoupling and cancellation, and the final simple and effective model by contrasting MLP and GCN views is merely a natural choice from the fundamental principle. **The conceptual contribution is recognized by both reviewers fKkq and EBbx in their initial reviews**. For example, reviewer fKkq remarked that ``The core insight of decoupling and contrasting "feature noise" and "structural noise" is both elegant and intuitive. It provides a principled argument against the need for complex, heuristic-based data augmentations, representing a significant departure from mainstream GCL research.'' **Therefore, the difference between our approach and SimMLP and other mainstream GCL already differs at the conceptual level**.

Furthermore, we have addressed the concerns raised by Reviewer pexd regarding ablation studies and large-scale dataset evaluation. We clarify that **both aspects have already been examined on large-scale heterophilic datasets in the original manuscript**, and we have now explicitly directed the reviewer to the exact locations where these results appear. Reviewer pexd acknowledged our clarification and noted on **04:03, 26 Nov** that he was willing to **raise his score**. Although we have provided a clear explanation of our novelty and contributions, it appears that Reviewer pexd still has residual concerns about novelty, despite **not specifying which aspects he considers insufficiently novel.** Despite this, **we believe that Reviewer pexd may still misunderstand the novelty, significance, and contribution of our work. We emphasize that our core intention is to let GCL return to its essence—simplicity guided by a principled noise-cancellation viewpoint. This represents a fundamentally new perspective and a meaningful departure from mainstream GCL approaches, as highlighted by Reviewers fKkq and EBbx.**

In summary, we believe we have comprehensively addressed all reviewers’ concerns, including clarifying our novelty and contributions, strengthening the theoretical exposition, adding extensive evaluations across diverse benchmarks, and even demonstrating broader applicability. We remain confident that **the key strength of this work lies in uncovering the core essence of GCL from a fundamentally new perspective and providing a minimal yet effective instantiation of this principle, reflecting the scientific spirit that returning to essence and simplicity can be exceptionally powerful**.

We sincerely thank the AC for their time and effort on this work, and we kindly hope the AC may reconsider the reviewers’ assessments after reviewing our clarification for the final decision. 💐

Best regards,


Authors

---

> ### Author Response · Authors · 2025-11-29
> **Response to Reviewer UouA's remaining two concerns (Q1)**
>
> For your reference, we post our response to Reviewer UouA's remaining two concerns as below:
>
> **Q1. Validation of adversarial robustness and graph-level classification performance limited to two datasets; require additional benchmarks. (at least 4 diverse datasets)**
>
> Following your suggestions, we have added additional standard benchmarks (covering up to four diverse datasets) for both adversarial robustness and graph-level classification. **These new results are reported in Table 7 (line 503) and Table 11 (line 1173 in the Appendix) of the revised manuscript, respectively**. For convenience, we also relist the corresponding tables below.
>
> Regarding **Table 1 (white-box adversarial attacks)**, the updated results clearly demonstrate the **strong robustness of our method**, particularly under the more challenging setting where both node attributes and the graph topology are jointly perturbed. This robustness is especially evident on heterophilic graphs, where structural perturbations are generally more harmful.
>
> For **Table 2 (graph-level classification)**, we **have added two additional standard benchmarks (PTC-MR and MUTAG) and also reported the average ranking** computed as the arithmetic mean of the valid per-dataset ranks (excluding missing entries). Although our method is not the top performer on every dataset, it achieves the lowest overall average rank, indicating the **best overall performance across all benchmarks**. Though not specifically designed for graph classification, GCN-MLP delivers competitive results compared with node-level baselines and performs on par with specialized graph-level contrastive methods, demonstrating that the proposed framework generalizes effectively beyond node-level tasks.
>
>  **Table 1: White-box attack robust accuracy (\%) under 15\% perturbation for node classification.**
>
> | Dataset | Attack | FROND| Res-GRACE | GraphACL | PolyGCL | LOHA   | EPAGCL   | SDMG    | GCN-MLP   |
> |-|-|-|-|-|-|-|-|-|-|
> | Photo| clean| 92.03±1.27 | 92.50±0.17 | 93.31±0.19 | 91.45±0.35 | 86.46±0.41 | 93.05±0.23 | 94.10±0.20 | **93.41±0.88** |
> |          | PGD |14.18±5.16 | 45.46±2.05 | 30.94±3.62 | 22.93±2.73 | 59.35±0.43 | 7.71±1.41 | 3.11±1.11 | **90.56±0.54** |
> |   | PRBCD |15.08±7.52|40.72±3.70 | 28.56±1.16 |15.86±1.99  | 52.82±3.45 | 3.69±1.35  | 21.35±11.31 | **75.42±0.69**|
> | Texas | clean| 74.86±3.21| 54.59±5.51 | 71.08±0.34 | 72.43±4.86 | 69.73±6.26| 68.92±4.05 | 53.60±2.67 |**78.38±4.68**|
> | | PGD| 69.46±7.16 | 28.85±8.55| 21.62±7.20 | 56.76±16.22 |32.70±20.17 | 48.65±17.72 | 27.57±22.67 | **75.68±6.28** |
> | | PRBCD | 65.14±3.91| 26.49±11.22| 23.78±8.99| 53.51±13.23| 37.57±15.17| 46.22±16.24| 15.68±16.12| **67.03±5.10** |
> | Wisconsin | clean| 67.84±3.84| 52.35±7.18| 69.22±0.40|76.08±3.33| 76.05±6.08| 63.73±3.95| 52.68±1.21|**85.10±2.35**|
>  | | PGD| 62.16±6.01| 30.45±6.89| 8.04±3.75 | 66.27±6.06| 66.86±7.52| 18.43±10.57| 15.29±17.60| **80.98±4.64**|
>  || PRBCD | 57.06±5.71| 27.56±8.36 | 11.37±3.56| 60.59±5.65| 57.06±7.36 | 33.92±11.20 | 18.24±12.15 | **73.53±4.04**|
> | Cornell | clean| 63.24±9.38| 51.08±5.19| 59.33±1.48| 43.78±3.51| 54.05±7.05 | 52.97±5.82| 45.59±0.67 | **73.78±5.68**|
>  || PGD| 44.36±8.34| 30.58±6.13| 29.46±10.23 | 38.65±6.17 | 48.11±5.10 | 20.00±9.98 | 39.46±11.67 | **53.78±8.92** |
> || PRBCD| 43.81±8.00| 28.13±5.47| 37.03±8.74| 36.76±7.47| 44.05±7.65 | 20.54±13.20 | 13.24±8.24| **52.16±9.52** |
>
> **Table 2: Graph classification results (%). The first 4 rows are from node-level GCL methods adapted to graph-level tasks, and the next 3 rows are from graph-level models.**
>
> | Method | Proteins | DD| PTC-MR | MUTAG | Avg. rank |
> |-|-|-|-|-|-|
> | MVGRL | 74.02±0.30 | 75.20±0.40 | - - | 89.20±1.30     | 5.33      |
> | GraphACL| 73.50±0.70 | - -| - - | 89.40±2.00     | 5.50      |
> |   SimMLP    | 75.30±0.10       | 78.40±0.50      | 60.30±1.10     | 87.70±0.20     | 3.63      |
> |   SDMG      | 73.16±0.16       | 72.66±3.16      | 56.70±2.02     | 91.58±0.28     | 5.00      |
> |   InfoGraph | 74.44±0.40       | 72.85±1.70      | 61.70±1.40     | 89.00±1.10     | 4.50      |
> |  GraphCL   | 74.39±0.45       | 78.62±0.40      | - -            | 86.80±1.30     | 4.67      |
> |   DRGCL     | 75.20±0.60       | 78.40±0.70      | - -            | 89.50±0.60     | 2.83      |
> | **GCN-MLP** | **75.41±0.35**   | **77.00±0.45** | **62.27±1.44**  | **89.56±0.85**  | **2.00** |

---

> ### Author Response · Authors · 2025-11-29
> **Response to Reviewer UouA's remaining two concerns (Q2)**
>
> **Q2. Difference between ours and SimMLP and related contrastive works & new experiments on addtional graphs**
>
> We would like to clarify that our contrastive formulation is **fundamentally different from SimMLP and mainstream GCL methods such as PolyGCL and GraphACL**. As stated in the original SimMLP paper (See line 14 in the abstract), SimMLP’s core objective is to inject structural information into MLPs by aligning a graph-context-aware GNN with a neighborhood-independent MLP. **Its purpose is therefore architectural enhancement of MLPs, rather than noise modeling**.
>
>
> In contrast, we formulate GCL from a **noise-cancellation perspective**: feature noise and structural noise are weakly correlated, enabling stronger mitigation when contrasted and combined. **To the best of our knowledge, this novel perspective is not explored in GCL literature**. We **theoretically establish this principle in Observation 1 and Theorem 1 (Appendix B)**. Their weak correlation ensures that contrastive interaction (and simple linear fusion) achieves stronger noise mitigation. This insight follows a basic vector-analysis principle: the less correlated two vectors are, the more effectively their noise cancels when combined. **To our knowledge, ours is the first work to explicitly frame GCL from this noise-cancellation viewpoint, rather than from a multi-view augmentation paradigm. As Reviewer fKkq noted, this represents a significant departure from mainstream GCL research.**
>
> Mainstream GCL methods (e.g., PolyGCL and GraphACL) rely on constructing multiple graph views (typically local views and global views) using polynomial filters or one-/two-hop neighborhood contexts. **Our goal is different: we intentionally avoid complex view construction and instead return to a simple, principled mechanism driven by noise-cancellation**. The resulting GCN-MLP architecture provides a minimal yet effective instantiation of this principle: the GCN captures structural signal plus structural noise, while the MLP isolates feature-level variations. Their weak correlation makes the contrastive interaction particularly effective, yielding strong empirical performance despite the lightweight design.We believe this reflects the modern notion that **simplicity can be exceptionally powerful**.
>
> Regarding the reviewer’s concern about DD, we acknowledge that GCN-MLP does not outperform SimMLP on this specific dataset. **However, as shown in Table 2, our method achieves superior results on three other standard graph-classification benchmarks (Proteins, PTC-MR, MUTAG)**. For node classification, GCN-MLP shows notably strong performance on heterophilic graphs (e.g., Wisconsin and Roman), where SimMLP typically struggles (Table 3), while also requiring less training time and comparable memory usage. **These additional evaluations provide a broader picture and confirm the effectiveness and efficiency of our proposed framework beyond any single dataset.**
>
>
>
> **Table 3: Accuracy / Training time (s) / Memory usage (MiB) comparison.**
>
> | Model     | Wisconsin                 | Roman                     |
> |-----------|----------------------------|----------------------------|
> | SimMLP    | 56.57 ± 2.59 / 0.008 / 79  | 64.72 ± 0.07 / 0.035 / 1159 |
> | **GCN-MLP**   | **85.29 ± 2.19 / 0.007 / 588** | **78.21 ± 0.39 / 0.097 / 5850** |

---

### Note · Authors · 2026-01-26

I have read and agree with the venue's withdrawal policy on behalf of myself and my co-authors.

---

### Meta-Review · Area_Chair_uMR3 · 2025-12-09

**Summary:**

Two of the four reviewers pointed out the limited novelty, confusing experimental settings, and the trade-off of experimental results. Besides, reviewer EBbx, who provides a weak accept score, possesses low confidence. By considering that Reviewer UouA and Reviewer pexd also have concerns on novelty and the trade-off of experimental results, I agree with Reviewer pexd that ‘the paper is marginally below the acceptance threshold’.

**Reviewer Concerns:**

Reviewer UouA and Reviewer pexd provided negative initial scores by considering novelty (similar to SimMLP and DGCL), overclaimed statements, experimental validation (missing task, robustness settings,  size of datasets,  and ablation study). Some of them are properly answered by providing additional experiments. However, both of them have concerns about novelty. Reviewer pexd clearly showed that ‘the paper is marginally below the acceptance threshold’. Reviewer fKkq also pointed out the weakness of performance on homophilic graphs.

**Reviewer Scores:**

Reviewer fKkq made it clear that he would maintain the previous scores after rebuttal.  Reviewer UouA and Reviewer pexd raised their score from 2 (reject) to weak reject (4). I think they will maintain these scores by considering the novelty. Based on these, I think the final scores may be as follows.
- EBbx	Rating: 6 / Confidence: 2
- pexd	Rating: 4 / Confidence: 4
- fKkq	Rating: 6 / Confidence: 4
- UouA	Rating: 4 / Confidence: 5

---

### Decision · Program_Chairs · 2026-01-26

Reject